# ZipIt! Merging Models from Different Tasks without Training

**George Stoica**[*]   **Daniel Bolya**[*]
**Jakob Bjorner**   **Pratik Ramesh**   **Taylor Hearn**   **Judy Hoffman**

Georgia Tech

{gstoica3,dbolya,jbjorner3,pramesh39,thearn6,judy}@gatech.edu

## ABSTRACT

Typical deep visual recognition models are capable of performing the one task they were trained on. In this paper, we tackle the extremely difficult problem of combining distinct models with different initializations, each solving a separate task, into one multi-task model **without any additional training**. Prior work in model merging permutes one model to the space of the other then averages them together. While this works for models trained on the same task, we find that this fails to account for the differences in models trained on disjoint tasks. Thus, we introduce "ZipIt!", a general method for merging two arbitrary models of the same architecture that incorporates two simple strategies. First, in order to account for features that aren't shared between models, we expand the model merging problem to allow for merging features *within* each model by defining a general "zip" operation. Second, we add support for *partially zipping* the models up until a specified layer, naturally creating a multi-head model. We find that these two changes combined account for 20-60% improvement over prior work, making it more feasible to merge models trained on disjoint tasks *without retraining*.

## 1   INTRODUCTION

Ever since AlexNet (Krizhevsky et al., 2017) popularized deep learning in computer vision, the field has thrived under the reign of massive models with an ever increasing number of parameters. Many vision problems once considered difficult or impossible are now benchmark tasks: classification with tens of thousands of classes (Deng et al., 2009; Zhou et al., 2017; Gemmeke et al., 2017), fast instance segmentation (He et al., 2017; Bolya et al., 2019), realistic image generation (Karras et al., 2018; Ho et al., 2020; Rombach et al., 2022), and more.

There are an abundance of independent, carefully tuned models out there for many tasks. However, if we want to expand an existing model's capabilities, we run into many potential issues. If we try training the model on an additional task, we face catastrophic forgetting (Kirkpatrick et al., 2017; Li & Hoiem, 2017; De Lange et al., 2021). If we evaluate the same model on different data without adaptation, we often find it doesn't generalize to out of domain samples (Blanchard et al., 2011; Muandet et al., 2013; Wang et al., 2022). We can try so called "intervention" strategies (Wang et al., 2022; De Lange et al., 2021) to mitigate these effects, but these often require further training which can be expensive (Dosovitskiy et al., 2020; Zhai et al., 2022; Dehghani et al., 2023). Instead, it would be nice if we could expand a model's capacity to solve new tasks by simply "zipping" it with other models trained on those tasks *without additional training*.

Combining multiple models into one has recently started to gain traction in the vision community. Model Soups (Wortsman et al., 2022a) can add multiple models finetuned from the same pretrained initialization to improve accuracy and robustness. Git Re-Basin (Ainsworth et al., 2022) generalizes further to models trained on the same data but with different initializations, though with a significant accuracy drop. REPAIR (Jordan et al., 2022) improves on Git Re-Basin by adding new parameters and adjusting model batch norms where applicable. However, all of these methods only combine models trained on the same task. In this paper, we take this line of work to a logical extreme: merging differently initialized models trained on *completely separate* tasks (see Fig. 1ab). We show that this is an incredibly difficult problem for prior work and employ two simple strategies to make it feasible.

---

[*]Equal Contribution. Code: https://github.com/gstoica27/ZipIt.

Figure 1: **Setting and ZipIt!** (a) Prior work merges differently initialized models from the **same dataset** with the **same** label sets: e.g., merging two models both trained to classify dog breeds. (b) Our setting expands this to merging models from **different** datasets with **different** label sets: e.g., merging a model that classifies dog breeds with one that classifies bird species. (c) **ZipIt!** merges these models *without retraining* by identifying shared features.

First, we note that prior work focuses on *permuting* one model to the other when merging them. This creates a 1-1 mapping between the two models, inherently assuming that most features *across* them are correlated. Since this isn't necessarily the case for models trained on different tasks, we cannot rely on permutation alone. Instead, we generalize model merging to support "zipping" any combination of correlated features *within* and *across* each model. We find that on some tasks, this alone improves accuracy **by up to 20%** vs. permutation-based approaches. Moreover, we prove that merging within models can yield a better result in the theoretical setting of Entezari et al. (2021).

Second, existing methods merge *the entire network*. While this might work for extremely similar models trained in the same setting, the features of models trained on disjoint tasks become less correlated over the course of the network (Kornblith et al., 2019). To solve this, we introduce *partial zipping*, where we only "zip" up to a specified layer. Afterward, we feed the merged model's outputs to the remaining unmerged layers of the original networks, creating a multi-head model. Depending on task difficulty, this can improve accuracy **by over 15%** while still keeping most layers merged.

Incorporating both of these strategies, we introduce ZipIt! (Fig. 1c), a general method for "zipping" any number of models trained on different tasks into a single multitask model *without retraining*. By deriving a general graph-based algorithm for merging and unmerging (Sec. 4), we can zip models of the same architecture together, merge features *within* each model, and partially zip them to create a multi-task model. We validate our approach by merging models trained on entirely disjoint sets of CIFAR (Krizhevsky et al., 2009) and ImageNet (Deng et al., 2009) categories, as well as merging several models trained on completely independent datasets into one, significantly outperforming prior work (Sec. 5). Finally, we ablate and analyze our method's capabilities on these scenarios (Sec. 6).

## 2 RELATED WORK

Model merging combines the weights of two or more models into a one. Our work differs from prior work in that we adapt mode connectivity techniques to target models trained on disjoint tasks (Fig. 1).

**Merging Finetuned Models.** If two models are finetuned from the same pretrained checkpoint, they often lie in the same error basin (Neyshabur et al., 2020). Several works (Huang et al., 2017; Izmailov et al., 2018; Von Oswald et al., 2020; Wortsman et al., 2022b; Ilharco et al., 2022b; Don-Yehiya et al., 2023) have exploited this property to average together the weights of a model at different stages of training. Tarvainen & Valpola (2017); Cai et al. (2021); Grill et al. (2020); Caron et al. (2021); Baevski et al. (2022) use an "exponential moving average" of training checkpoints as a teacher for self-supervised learning. Other works merge models initialized from the same pretrained base, but that were finetuned independently, either by simply averaging their weights (McMahan et al., 2017; Wortsman et al., 2022a; Choshen et al., 2022; Ramé et al., 2022), permuting one model to the other (Ashmore & Gashler, 2015; Yurochkin et al., 2019; Wang et al., 2020), combining meaningful weight regions (Ilharco et al., 2022a; Gueta et al., 2023; Yadav et al., 2023; Sung et al., 2023), or maximizing an objective (Matena & Raffel, 2021). Our setting differs, as we do not assume the same initialization.

**Merging Differently Initialized Models.** Merging models with different initializations is a much more challenging problem. Works in this space often rely on *mode connectivity* (Freeman & Bruna, 2016; Garipov et al., 2018; Draxler et al., 2018; Frankle et al., 2020), attempting to interpolate between models along a low loss path (e.g., Tatro et al. (2020); Singh & Jaggi (2020); Liu et al. (2022)). Most recent work follow the intuition, later formalized by Entezari et al. (2021), that

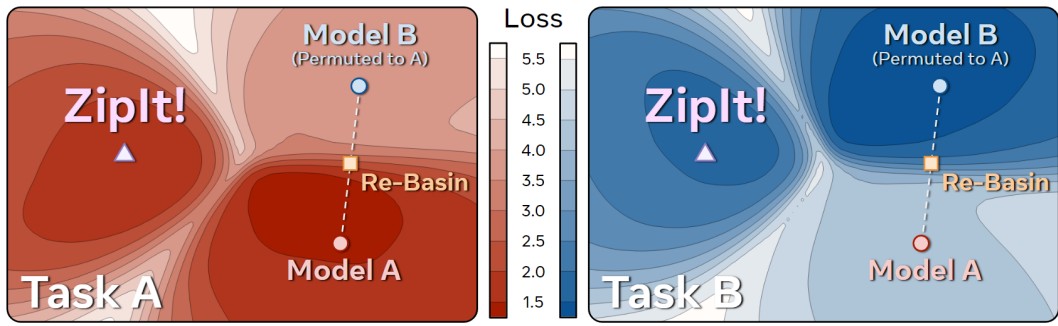

Figure 2: **Task Loss Landscapes** for models in Tab. 1b. Model A and Model B lie in low loss basins for their own tasks, but *not for the other task*. Thus, any interpolation between Model A and a permuted Model B (e.g., Git Re-basin) lies outside the minima *for both tasks* and thus performs poorly. In contrast, ZipIt! improves the merge by finding a model that lies in a low loss basin for both.

models permuted to the same loss basin can be merged by averaging their weights. Most notably, Git Re-Basin (Ainsworth et al., 2022) permutes models by comparing the similarity between their weights. REPAIR (Jordan et al., 2022) improves the accuracy of Git Re-Basin by instead computing the correlation between their intermediate layer feature activations, and adding several batch norms to the network. Peña et al. (2022) find permutations using global rather than local optimization, though they don't support skip connections. Some of these works (e.g., Singh & Jaggi (2020); Ainsworth et al. (2022)) evaluate on on a setting where each model sees varying numbers of instances per class. And Peña et al. (2022) evaluates on a continual learning setting with disjoint categories, but their method requires training optimization. Similarly, He et al. (2018) merges models of different tasks, but requires jointly finetuning after each layer merge. As far as we are aware, we present the first *general method* to successfully merge models trained on disjoint tasks *without additional training*.

## 3 BACKGROUND AND MOTIVATION

Model merging stems from mode connectivity (Garipov et al., 2018), where it is conjectured that models trained with SGD on the same dataset lying in the same *loss basin* (i.e., region or *mode* of low loss) can be combined into a single model that's just as performant as the original models. If these models can be combined well by linearly interpolating their weights, they are said to be *linearly mode connected* (LMC) (Garipov et al., 2018; Entezari et al., 2021). Our work is similar to finding LMC, but across *different datasets with disjoint label sets* (i.e., separate tasks as in De Lange et al. (2021)).

Consider a model $\mathcal{L}$ as a collection of layers $L_i \in \mathcal{L}$, each of which may have some parameters (e.g., $W_i, b_i$ for a linear layer). If $\mathcal{L}^A$ and $\mathcal{L}^B$ are finetuned from the same checkpoint, several works (e.g., Izmailov et al. (2018); Wortsman et al. (2022a)) find merging them is as easy as linearly interpolating their weights (i.e., they are LMC). E.g., if $L_i$ is a linear layer, the new weight matrix $W_i^*$ is simply

$$W_i^* = \gamma W_i^A + (1 - \gamma) W_i^B \tag{1}$$

with an interpolation constant $\gamma \in [0, 1]$, usually set to $1/2$. However, if $\mathcal{L}^A$ and $\mathcal{L}^B$ were not finetuned from the same checkpoint, they often do not lie in the same mode (Entezari et al., 2021; Ainsworth et al., 2022) and cannot be interpolated. Indeed, Eq. 1 typically results in random accuracy.

To fix this, Entezari et al. (2021) conjecture that *large enough* models are likely LMC modulo permutation. This is because (1) many neural networks can be permuted internally without affecting their outputs and (2) permutation can move two models into the same basin, allowing much of the lost accuracy to be recovered. More concretely, let $P_i$ be a permutation matrix that permutes outputs of layer $L_i^B$ to the space of $L_i^A$. Then for each layer, permutation works apply

$$W_i^* = \gamma W_i^A + (1 - \gamma) P_i W_i^B P_{i-1}^T \tag{2}$$

Note that here we permute the output space of $W_i^B$, but we also need to permute its input space to undo the permutation from the previous layer (hence the use of $P_{i-1}^T$).

**Problems with Permutation.** Eq. 2 relies on the likelihood of model B lying in the same basin as model A after permutation being high. However, this is far less likely when the models are trained

on different tasks, as each model optimizes for basins containing distinct task-specific information. In this case the optimal permutation of model B to model A still lies in a strong basin on task B but *doesn't* lie in a basin on task A, as shown in Figure 2. This causes the interpolated model to perform worse than either the two original models. Thus, we explore alternative merging methods.

## 4    ZIPIT!

In this work, we treat model merging as combining the checkpoints (i.e., collection of weights) of multiple models into a single checkpoint that can perform all the tasks of its constituents. We do this by merging the layers of the models together. For instance, suppose $L_i \in \mathcal{L}$ is a linear layer with parameters $W_i \in \mathbb{R}^{n_i \times m_i}, b_i \in \mathbb{R}^{n_i}$ with input features $x \in \mathbb{R}^{m_i}$ and outputs features $f_i \in \mathbb{R}^{n_i}$:

$$f_i = L_i(x) = W_i x + b_i \tag{3}$$

Our goal is to take $L_i^A \in \mathcal{L}^A$ from model A and $L_i^B \in \mathcal{L}^B$ from model B and merge them into a layer $L_i^*$ that combines their feature spaces such that information from both $f_i^A$ and $f_i^B$ is retained in $f_i^*$. We accomplish this by merging each layer of one model with the corresponding layer in the other, both merging features in one *across* both layers or *within* the same layer. This is in contrast to permutation-based merging method, which only combine features *across* layers.

**Why should we merge *within*?** Features of models trained on different tasks may be dissimilar, as the models solve different problems. Forcibly combining these dissimilar features can yield merges that don't perform well on either original task (Fig 2). Instead, those features may be more compatible with others within the same model, which would better retain performance when combined.

In fact, we can *prove* that methods which allow merges *within* each model (as well as across both) perform equal to or *better* than those which only merge across models (e.g., permutation-reliant approaches) in a limited but prevalent setting. Specifically, we obtain a tighter bound over Theorem 3.1 from Entezari et al. (2021) when redundancy exists within a model and is leveraged. Both Theorem 3.1 and our Theorem 1 (see Appendix G for formalization and proof) bound the degree to which the loss of a merged two-layer model with $d$-input dimensions and $h$-intermediate dimensions increases compared to the losses of the original models. Theorem 3.1 bounds this increase to $\tilde{O}(h^{-1/(2d+4)})$. However, if features within a model are *redundant*, then we reduce the bound to

$$\text{Loss Increase of Merged Model} \leq \begin{cases} \tilde{O}\left(\left(\frac{h}{1-2\Gamma}\right)^{-\frac{1}{2d+4}}\right) & \Gamma < 0.5 \\ 0 & \text{otherwise} \end{cases} \tag{4}$$

with $\Gamma \in [0, 1]$ measuring what portion of features are redundant. This bound is $\sqrt[2d+4]{1-2\Gamma} \leq 1$ times that of Theorem 3.1 when $\Gamma < 0.5$ (equal only when $\Gamma = 0$) and *explicitly zero* when $\Gamma \geq 0.5$.

**How do we merge features?** In prior work, each of the merged features $f_i^*$ is the result of combining one feature from $f_i^A$ and one from $f_i^B$. However in our case, we can also merge features by combining two from just $f_i^A$ or two from just $f_i^B$. To account for this, we concatenate $f_i^A$ and $f_i^B$ into a single feature vector: $f_i^A \| f_i^B \in \mathbb{R}^{2n_i}$. Then, like prior work (e.g. Li et al. (2015); Jordan et al. (2022)), we define feature similarity as the pairwise correlation between between neuron activations over a small set of images (without labels). However, unlike those works, we compute correlations between every activation in *the full concatenated space* $f_i^A \| f_i^B$. Our approach thus measures the similarity of every feature $f_i^A$ and $f_i^B$ to all features in both models, rather than solely between $f_i^A$ and $f_i^B$.

Next, if two features are well correlated, we can average them without losing much information. Thus, we can construct $f_i^*$ by finding $n_i$ pairs of similar features in $f_i^A \| f_i^B$ and averaging them together. By default, we do this greedily: i.e., iteratively match the features with the highest correlation without replacement; though we explore extensions to this in Sec. 4.3 and test other methods in Tab. 4. Then we can use these matches to construct $f_i^*$. Formally, we define a "merge matrix" $M_i \in \mathbb{R}^{n_i \times 2n_i}$ s.t.

$$f_i^* = M_i\left(f_i^A \| f_i^B\right) \tag{5}$$

$M_i$ averages the matched features, with each match corresponding to one output feature in $f_i^*$. For instance, if $u$th match is between indices $s, t \in \{1, \ldots, 2n_i\}$ of $f_i^A \| f_i^B$, then the $u$th row of $M_i$ would be $1/2$ at columns $s$ and $t$ and 0 elsewhere. This results in $f_i^*[u] = \frac{1}{2}(f_i^A \| f_i^B)[s] + \frac{1}{2}(f_i^A \| f_i^B)[t]$.

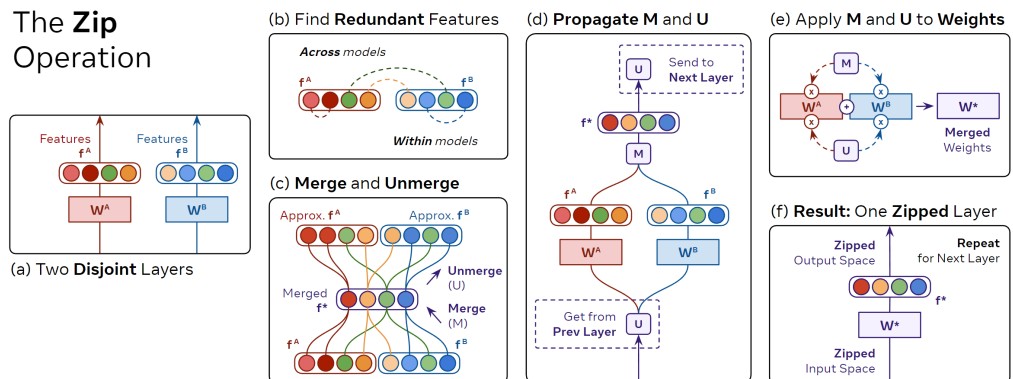

Figure 3: **ZipIt!** merges models layer-wise by exploiting redundancy in their features. (a) Output features $f^A$ and $f^B$ from two disjoint layers are (b) paired with other features based on the similarity of their activations. (c) We produce a merge matrix $M$ to combine the pairs into a single shared output feature space, and a corresponding unmerge matrix $U$ that undoes this operation. (d) We then propagate $U$ up the network to align the next layer's input space, and simultaneously receive the previous layer's $U$ to align our input space. (e) We apply Eq. 8 to "zip" the layers together using the $M$ for the output and $U$ for the input, producing a single layer (f). We then repeat (a) on the next layer.

Thus, applying $M_i$ has the effect of interpolating with $\gamma = 1/2$ but is more general (e.g., allows for merging more than 2 models at once, see Sec. 4.3).

**What about the next layer?** After merging features in one layer, we now have the problem that the next layers, $L_{i+1}^A, L_{i+1}^B$, are incompatible with $f_i^*$. Instead, we need to *undo* the merge operation before passing the features to the next layer. Thus, we define an "unmerge" matrix $U_i \in \mathbb{R}^{2n_i \times n_i}$ s.t.

$$U_i f_i^* \approx f_i^A \| f_i^B \tag{6}$$

$U_i$ is the pseudoinverse of $M_i$ and in the case of the matching from earlier is simply $2M_i^T$. Note that strict equality is unlikely here. Like in prior work, merging models is a lossy operation.

We split this unmerge matrix in half along its rows into $U_i^A, U_i^B \in \mathbb{R}^{n_i \times n_i}$ that act individually to produce $f_i^A$ and $f_i^B$. With this, we can evaluate the next layers using the merged features:

$$f_{i+1}^A \approx L_{i+1}^A(U_i^A f_i^*) \qquad f_{i+1}^B \approx L_{i+1}^B(U_i^B f_i^*) \tag{7}$$

### 4.1 THE "ZIP" OPERATION

We now have all the necessary pieces, and can derive a general operation to merge $L_i^A$ and $L_i^B$ at an arbitrary point in the network (Fig. 3). First, we compute $M_i$ and $U_i$ by matching features between $f_i^A$ and $f_i^B$. We then pass $U_i$ to the next layer and receive $U_{i-1}$ from the previous layer. Using $M_i$ and $U_{i-1}$, we "fuse" the merge and unmerge operations into the layer's parameters. For a linear layer:

$$W_i^* = M_i^A W_i^A U_{i-1}^A + M_i^B W_i^B U_{i-1}^B \tag{8}$$

where $M_i^A$ and $M_i^B$ are $M_i$ split along its columns. $b_i^*$ has the same equation but without unmerging.

Note the similarity between Eq. 8 and Eq. 2. This isn't a coincidence: if we only allowed merging *across* models and not *within* models, our "zip" operation would be identical to Git Re-Basin's permute-then-interpolate approach. Thus, Eq. 8 can be thought of as a generalization of prior work.

### 4.2 ZIP PROPAGATION

However, most modern neural networks are not simply collections of linear layers stacked on top of each other. In practice, we cannot combine merge and unmerge matrices into every layer of the network, as a local zip (Eq. 8) expects the layer to have a weight *matrix*—i.e., the layer has to have separate input and output spaces so that we can unmerge the input space and merge the output space. Other layers (e.g., BatchNorm, ReLU) don't have such a weight matrix.

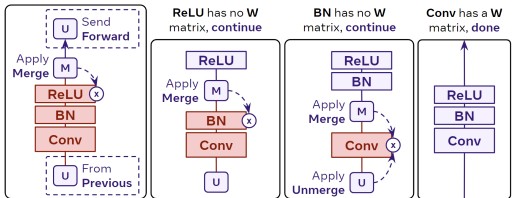

Figure 4: **Zip Propagation.** We propagate $M_i$ backward until we hit a layer with weights, merging merging element-wise layers (e.g., Batch-Norm) along the way.

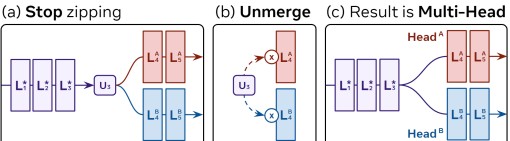

Figure 5: **Partial Zip.** (a) If we stop zipping early and (b) apply the latest U from the zip propagation to the inputs of the first unmerged layer in each model, (c) we get a multi-head model with a head for each task.

Thus, we "propogate" $M_i$ and $U_i$ *through* these layers. For instance, in Fig. 4, we show a common stack of layers found in a typical ConvNet. Following Jordan et al. (2022), we compute $M_i$ and $U_i$ using the activations of the network (i.e., after each ReLU). We can't fuse $M_i$ with the ReLU layer, as it doesn't have any parameters. Similarly, we can merge the parameters of the preceding BatchNorm layer (i.e., in the same way as bias). But it doesn't have a weight matrix, so we also can't fuse $M_i$ into it. Only once we've reached the Conv layer can we fuse $M_i$ and $U_i$ into it using Eq. 8 (in this case, treating each kernel element as independent).

Similar care needs to be taken with skip connections, as every layer that takes input from or outputs to a skip connection shares the same feature space. However, this too can be dealt with during propagation—we just need to propagate $M_i$ backward and $U_i$ forward to each layer connected by the same skip connection. In general, we can define propagation rules to handle many different types of network modules (see Appendix C).

### 4.3 EXTENSIONS

**Partial Zip.** We don't always want to zip every layer of the two networks, especially if their output spaces are incompatible, or if doing so would lose too much accuracy. Instead, we can perform a *partial zip*. That is, we zip most of the layers together, but leave the later ones *unzipped* (Fig. 5).

Implementing this operation is simple in our framework: zip as normal until the specified layer $i$, then the remaining unzipped layers will receive $U_i$ through zip propagation. If we apply $U_i^A$ to $L_{i+1}^A$ and $U_i^B$ to $L_{i+1}^B$, the remaining unzipped layers will form "heads" that expect merged features as input. We can then ensemble the heads or choose one to evaluate at runtime.

**Repeated Matching** ($\alpha$). In some cases, we'd like to merge more than two models together. To do this, we allow "repeated matches". That is, when two features are matched in our greedy algorithm, they are removed and replaced with the resulting merged feature. To ensure that one feature doesn't get merged endlessly, we set the correlations of the new feature to be the minimum of the old features' similarities weighted by $\alpha \in (0, 1]$. We find a small value of $\alpha$ typically works best.

**Same-model Budget** ($\beta$). To demonstrate the effectiveness of same-model merges, we introduce a "budget" parameter $\beta \in [0, 1]$ that denotes what percent of total merged features can come from models merging within themselves, with each model receiving an equal portion of this budget. Note that a budget of 0 results in Eq. 2, as in that case no features can be merged within models.

## 5 RESULTS

There is no standard benchmark to evaluate merging approaches on models from distinct tasks, so we construct our own. We evaluate our approach in two different settings. (1) A versatile test-bed: disjoint category splits of the same dataset (i.e., *same dataset and different label sets*). (2) A very challenging setting: completely different datasets and tasks (i.e., *different datasets and label sets*).

**Experimental Details.** For each experiment where we sample multiple disjoint splits of categories, we hold one split out for hyperparameter search and report mean and standard deviation on the rest. For experiments with models trained on different datasets, we subsample the validation set into a validation and test set to use for the same purpose. To compute correlations, we use a portion of the

|  |  | Accuracies (%) | | | |
| Method | FLOPs (G) | Joint | Task A | Task B | Avg |
| --- | --- | --- | --- | --- | --- |
| Model A | 0.68 | $48.2_{\pm1.0}$ | $97.0_{\pm0.6}$ | $45.1_{\pm8.6}$ | $71.0_{\pm4.4}$ |
| Model B | 0.68 | $48.4_{\pm3.8}$ | $49.1_{\pm9.3}$ | $96.1_{\pm1.1}$ | $72.6_{\pm4.9}$ |
| W. Avg (Eq. 1) | 0.68 | $43.0_{\pm1.6}$ | $54.1_{\pm1.4}$ | $67.5_{\pm1.2}$ | $60.8_{\pm4.5}$ |
| Git Re-Basin‡ | 0.68 | $46.2_{\pm0.8}$ | $76.8_{\pm8.9}$ | $82.7_{\pm5.1}$ | $79.8_{\pm6.5}$ |
| Permute (Eq. 2) | 0.68 | $58.4_{\pm6.8}$ | $86.6_{\pm2.1}$ | $87.4_{\pm1.1}$ | $87.4_{\pm1.4}$ |
| **ZipIt!**$_{20/20}$ | 0.68 | $\mathbf{79.1}_{\pm1.1}$ | $\mathbf{92.9}_{\pm1.1}$ | $\mathbf{91.2}_{\pm1.4}$ | $\mathbf{92.1}_{\pm1.0}$ |
| Ensemble | 1.37 | $87.4_{\pm2.6}$ | $97.0_{\pm0.6}$ | $96.1_{\pm1.1}$ | $96.6_{\pm0.6}$ |
| **ZipIt!**$_{13/20}$ | 0.91 | $\mathbf{83.8}_{\pm3.1}$ | $\mathbf{95.1}_{\pm0.7}$ | $\mathbf{94.1}_{\pm1.5}$ | $\mathbf{94.6}_{\pm0.6}$ |

(a) **CIFAR-10 (5+5).** ResNet-20 (4× width).

|  |  | Accuracies (%) | | | |
| Method | FLOPs (G) | Joint | Task A | Task B | Avg |
| --- | --- | --- | --- | --- | --- |
| Model A | 2.72 | $41.6_{\pm0.3}$ | $82.9_{\pm0.7}$ | $24.8_{\pm0.4}$ | $53.9_{\pm0.5}$ |
| Model B | 2.72 | $41.6_{\pm0.2}$ | $25.1_{\pm1.2}$ | $82.8_{\pm0.2}$ | $54.0_{\pm0.6}$ |
| W. Avg (Eq. 1) | 2.72 | $17.0_{\pm1.7}$ | $23.8_{\pm6.9}$ | $24.8_{\pm5.9}$ | $24.3_{\pm1.9}$ |
| Git Re-Basin‡ | 2.72 | $40.9_{\pm0.2}$ | $57.3_{\pm1.5}$ | $56.7_{\pm0.7}$ | $57.0_{\pm0.8}$ |
| Permute (Eq. 2) | 2.72 | $42.8_{\pm0.7}$ | $61.6_{\pm1.4}$ | $60.5_{\pm0.5}$ | $61.0_{\pm0.8}$ |
| **ZipIt!**$_{20/20}$ | 2.72 | $\mathbf{54.9}_{\pm0.8}$ | $\mathbf{68.2}_{\pm0.8}$ | $\mathbf{67.9}_{\pm0.6}$ | $\mathbf{68.0}_{\pm0.4}$ |
| Ensemble | 5.45 | $73.5_{\pm0.4}$ | $82.9_{\pm0.7}$ | $82.8_{\pm0.2}$ | $82.8_{\pm0.4}$ |
| **ZipIt!**$_{13/20}$ | 3.63 | $\mathbf{70.2}_{\pm0.4}$ | $\mathbf{80.3}_{\pm0.8}$ | $\mathbf{80.1}_{\pm0.7}$ | $\mathbf{80.2}_{\pm0.6}$ |

(b) **CIFAR-100 (50+50).** ResNet-20 (8× width).

Table 1: **CIFAR Results.** ZipIt! vs. baselines on combining a model trained on half the classes (Task A) with one trained on the other half (Task B) *without extra training*. We report both joint (10/100-way) and per-task (5/50-way) accuracy. ZipIt! *significantly* outperforms its baseline and closes in on the upper bound (ensemble accuracy). ‡ refers to Ainsworth et al. (2022).

training set for each dataset as in Li et al. (2015) (see Appendix B). For a fair comparison, we reset the batch norms for *all* methods (including the original models) using the training data (following the recommendation in Jordan et al. (2022)). For our method, ZipIt!$_{n/m}$ indicates that $n$ out of the $m$ layers in the network have been zipped (Sec. 4.3). Note, all our models have *different initializations*.

**Evaluation.** For the setting with disjoint class splits of the same dataset, we evaluate performance in two ways: joint accuracy and per task accuracy. For joint accuracy, we evaluate each model over *all* classes in the combined dataset. For per task accuracy, we compute the accuracy of each task individually (i.e., supposing we had task labels at runtime) and then report the average. The former is similar to a continual learning setting where we want to augment the knowledge of the model, while the latter is akin to a multi-task setting where we know which task we're using at test time. For the scenario where we merge models trained on different datasets, we use the per task accuracy metric, as the label spaces are not comparable.

**Baselines.** In addition to the default Weight Matching version of Git Re-Basin (Ainsworth et al., 2022), we compare to two baselines: Weight Averaging (Eq. 1) and Permute (Eq. 2) with $\gamma = 1/2$ using our framework (i.e., we set $M_i$ and $U_i$ such that Eq. 8 is equivalent). For Permute, we use linear sum assignment to find optimal permutations (following Li et al. (2015)). Note that our Permute is a *strong* baseline we create using our framework and is more accurate than Git Re-Basin in our settings. It's also similar to REPAIR (Jordan et al., 2022), but without adding extra parameters to the model. Finally, with perfect merging, the merged model's outputs would be identical to the originals. Thus we include Ensemble as an upper bound (executing and concatenating the results of both models).

## 5.1 CIFAR-10 AND CIFAR-100

We train 5 pairs of ResNet-20 (He et al., 2016) from scratch with different initializations on disjoint halves of the CIFAR-10 and CIFAR-100 classes (Krizhevsky et al., 2009). While ZipIt! supports "partial zipping" to merge models with different outputs (in this case, disjoint label sets), prior methods without retraining do not. To make a fair comparison, we train these CIFAR models with a CLIP-style loss (Radford et al., 2021) using CLIP text encodings of the class names as targets. This way, both models output into the same CLIP-space regardless of the category. Note, this means the models are capable of some amount of zero-shot classification on the tasks they were not trained on.

**CIFAR-10 (5+5).** In Tab. 1a, we merge models trained on disjoint 5 class subsets of CIFAR-10 using ResNet-20 with a 4× width multiplier (denoted as ResNet-20×4). In joint classification (i.e., 10-way), Git Re-Basin is unable to perform better than using either of the original models alone, while our Permute baseline performs slightly better. In stark contrast, our ZipIt! performs a staggering *32.9%* better than Git Re-Basin and *20.7%* better than our baseline. If allow the last stage of the network to remain unzipped (i.e., zip up to 13 layers), our method obtains 83.8%, which is only 3.6% behind an ensemble of model A and model B (which is practically the upper bound for this setting). We also achieve similar results when merging VGG11 models in this setting (Appendix D).

**CIFAR-100 (50+50).** We find similar results on disjoint 50 class splits of CIFAR-100 in Tab. 1b, this time using an 8× width multiplier instead. Like with CIFAR-10, Git Re-Basin fails to outperform even the unmerged models themselves in joint classification (i.e., 100-way), and this time Permute is only 1.2% ahead. ZipIt! again *significantly* outperforms prior work with +14% accuracy over Git Re-Basin for all layers zipped, and a substantial +29.2% if zipping 13/20 layers. At this accuracy,

ZipIt!$_{13/20}$ is again only 3.3% behind the ensemble for joint accuracy and 2.6% behind for average per task accuracy, landing itself in an entirely different performance tier compared to prior work.

## 5.2 IMAGENET-1K (200+200)

To test our method on the *much harder* setting of large-scale data, we train 5 differently initialized ResNet-50 models with cross entropy loss on disjoint 200 class subsets of ImageNet-1k (Deng et al., 2009). To compare to prior work that doesn't support partial zipping, we initialize the models with capacity for all 1k classes, but only train each on their subset.

In Tab. 2 we show results on exhaustively merging pairs from the 5 models. To compute joint (i.e., 400-way) accuracy, we softmax over each task's classes individually (like in Ahn et al. (2021)), and take the argmax over the combined

| Method | FLOPs (G) | Accuracies (%) | | | |
|---|---|---|---|---|---|
| | | Joint | Task A | Task B | Avg |
| Model A | 4.11 | 37.2$_{\pm2.0}$ | 74.3$_{\pm4.0}$ | 0.5$_{\pm0.1}$ | 37.4$_{\pm2.0}$ |
| Model B | 4.11 | 35.3$_{\pm1.6}$ | 0.5$_{\pm0.1}$ | 70.5$_{\pm3.2}$ | 35.5$_{\pm1.6}$ |
| W. Avg (Eq. 1) | 4.11 | 0.3$_{\pm0.1}$ | 0.6$_{\pm0.1}$ | 0.7$_{\pm0.1}$ | 0.6$_{\pm0.1}$ |
| Git Re-Basin[‡] | 4.11 | 3.1$_{\pm1.2}$ | 5.3$_{\pm2.6}$ | 5.7$_{\pm2.4}$ | 5.5$_{\pm1.7}$ |
| Permute (Eq. 2) | 4.11 | 8.6$_{\pm5.8}$ | **10.1**$_{\pm4.4}$ | 15.3$_{\pm11.1}$ | 12.7$_{\pm7.7}$ |
| **ZipIt!**$_{50/50}$ | 4.11 | 8.6$_{\pm4.7}$ | **12.4**$_{\pm5.9}$ | **14.7**$_{\pm7.8}$ | **13.5**$_{\pm6.6}$ |
| Ensemble | 8.22 | 63.3$_{\pm4.9}$ | 74.3$_{\pm4.0}$ | 70.5$_{\pm3.2}$ | 72.4$_{\pm2.5}$ |
| **ZipIt!**$_{22/50}$ | 6.39 | 55.8$_{\pm4.1}$ | 65.9$_{\pm2.5}$ | 64.1$_{\pm3.0}$ | 65.0$_{\pm2.3}$ |
| **ZipIt!**$_{10/50}$ | 7.43 | **60.9**$_{\pm4.1}$ | **70.7**$_{\pm3.0}$ | **69.0**$_{\pm2.9}$ | **69.9**$_{\pm1.9}$ |

Table 2: **ImageNet-1k (200+200) Results.** Merging ResNet-50 models trained from scratch on disjoint 200 category subsets (Task A and B) of ImageNet-1k. Prior work performs poorly, but ZipIt! makes this task feasible. [‡]Ainsworth et al. (2022).

400 class vector. On this extremely difficult task, Git Re-Basin only obtains 3.1% for joint accuracy (with random accuracy being 0.25%). Both the Permute baseline and ZipIt! with all layers zipped perform better, but with each at 8.6%, are still clearly lacking. Note that we find the same-model merging budget $\beta$ to not matter for this set of models (see Fig. 6), which suggests that there's not a lot of redundant information *within* each model in this setting. Thus, ZipIt! chooses to merge mostly *across* models instead, performing similarly to the permute baseline. We find this same trend in CIFAR with smaller models (see Fig. 6), and may be an artifact of model capacity. The story changes when we increase the capacity of the merged model by partial zipping: ZipIt!$_{10/50}$ reaches close to upper bound ensemble accuracy *on this extremely difficult task*, while saving on FLOPs.

## 5.3 MULTI-DATASET MERGING

We now take our model merging framework one step further by merging differently initialized models trained on *completely separate datasets and tasks*. We present two settings: merging multiple classification datasets and merging semantic segmentation with image classification.

**Image Classification Datasets.** Merging ResNet-50 models trained on: Stanford Dogs (Khosla et al., 2011), Oxford Pets (Parkhi et al., 2012), CUB200 (Welinder et al., 2010), and NABirds (Van Horn et al., 2015). In Tab. 3, we show the average per task accuracy from exhaustively merging each pair and the much more difficult setting of merging all four at once. We report the accuracy of our baselines by applying them up until the last layer, but we can't compare to prior work as they don't support this setting. As in all our previous experiment we merge *without retraining*.

| Method | FLOPs (G) | Per-Task Accuracies (%) | | | | |
|---|---|---|---|---|---|---|
| | | SD | OP | CUB | NAB | Avg |
| *Merging Pairs* | | | | | | |
| W. Avg (Eq. 1) | 4.11 | 12.9 | 18.2 | 13.9 | 0.2 | 11.3 |
| Permute (Eq. 2) | 4.11 | 46.2 | 47.6 | 35.6 | **13.5** | 35.7 |
| **ZipIt!**$_{49/50}$ | 4.11 | **46.9** | **50.7** | **38.0** | 12.7 | **37.1** |
| Ensemble | 8.22 | 72.7 | 81.1 | 71.0 | 77.2 | 75.5 |
| **ZipIt!**$_{22/50}$ | 6.39 | 62.6 | 71.2 | 62.8 | 53.0 | 62.4 |
| **ZipIt!**$_{10/50}$ | 7.42 | 66.5 | 75.8 | 65.6 | 66.8 | 68.7 |
| *Merging All 4* | | | | | | |
| W. Avg (Eq. 1) | 4.12 | 0.8 | 3.0 | 0.6 | 0.3 | 1.2 |
| Permute (Eq. 2) | 4.12 | 15.7 | 26.1 | **14.0** | **5.3** | 15.3 |
| **ZipIt!**$_{49/50}$ | 4.12 | **21.1** | **33.3** | 8.6 | 3.9 | **16.8** |
| Ensemble | 16.4 | 72.7 | 81.2 | 71.0 | 77.2 | 75.5 |
| **ZipIt!**$_{22/50}$ | 11.0 | 50.2 | 55.9 | 44.0 | 32.0 | 45.5 |
| **ZipIt!**$_{10/50}$ | 14.1 | **63.5** | **70.8** | **63.7** | **63.1** | **65.3** |

Table 3: **Multi-Dataset Results.** Merging ResNet-50 models trained on *completely different datasets*: Stanford Dogs (SD), Oxford Pets (OP), CUB200 (CUB), and NABirds (NAB). We report average per-task accuracy over merging model pairs, and all four.

For pairs of models, ZipIt! slightly outperforms our permute baseline across all tasks and performs similarly when merging all 4 models at once. However, if we add capacity to the merged model through partial zipping, we perform up to 33% better on merging pairs and 50% better on merging all four models than the permute baseline. Partial zipping is a significant factor to obtain strong performance, especially with more than 2 models.

**Multiple Output Modalities.** In Appendix F, we combine across modalities by merging the ResNet-50 backbone of a DeeplabV3 (Chen et al., 2017) segmentation model with an ImageNet-1k classification model. The resulting combined model can perform both semantic segmentation and image classification. Even with half of layers merged, ZipIt! retains good performance on both tasks.

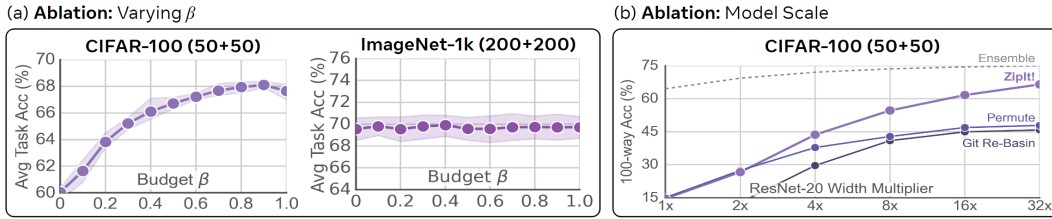

Figure 6: **Varying $\beta$ and Model Scale.** Left: We find when the model has enough capacity for the task, a high budget (Sec. 4.3) improves performance. Right: ZipIt! makes effective use of extra model capacity to quickly reach the ensemble on CIFAR-100 (50+50) when we increase the width of ResNet-20 models. In contrast, our baselines only slightly benefit from the extra scale.

## 6    ANALYSIS

**Merging *within* Models.** A critical piece of ZipIt! compared to prior work is the ability to merge *within* models, not just *across* models. In Sec. 4.3, we introduce a budget parameter $\beta$ to limit the number of same-model merges, and here use CIFAR-100 (50+50) and ImageNet-1k (200+200) to illustrate its effectiveness (Fig. 6a). On CIFAR, same-model merges are very important, with the optimal budget being above 0.8, meaning 80% of merges are allowed to be within the same model. This is not the case, however, on ImageNet, where the difficulty of the task means there likely are much fewer redundant features *within* each model.

**Model Scale.** In Fig. 6b, we test the effect of model scale directly by evaluating joint accuracy on our CIFAR-100 (50+50) setting with ResNet-20 models of increasing width. Here, we explicitly see that when the width of the models are too small for the task (e.g., $< 4\times$), ZipIt! and the Permute baseline perform identically (though both much better than Git Re-Basin). However, when the scale increases, ZipIt! trends toward the ensemble upper bound of 75%, while both the Permute baseline and Git Re-Basin plateau at around 45%. This corroborates Eq. 4 and indicates our method uses the extra model capacity effectively, much better than prior work.

**Matching Algorithm.** In Tab. 4, we compare matching algorithms used to compute $M_i$ in Eq. 8. Using either the identity (weight averaging) or a permutation (as in prior work) underperforms on CIFAR-10 (5+5) joint 10-way classification. In contrast, we obtain up to 21.2% higher accuracy if we allow both permutations and *merging within models*. However, doing this optimally is difficult, as the standard linear sum assignment algorithm assumes bipartite matches. We could use a optimal graph-based solver (e.g., Hagberg et al. (2008)) instead, but doing so is prohibitively slow (11 minutes to transform a ResNet-20×4 model). Thus, we find matches greedily by repeatedly taking the most corre-

| Algorithm | A↔A/B↔B? | Acc | Time |
|---|---|---|---|
| Identity (Eq. 1) | ✗ | $43.0_{\pm 3.1}$ | 1.8 ms |
| Permute (Eq. 2) | ✗ | $58.4_{\pm 1.3}$ | 28 ms |
| K-Means | ✓ | $29.1_{\pm 5.5}$ | 19 sec |
| *Zip (Eq. 8)* | | | |
| Optimal Match | ✓ | $\mathbf{79.6}_{\pm 1.7}$ | 11 min |
| Greedy Match | ✓ | $\mathbf{79.0}_{\pm 1.8}$ | 1.1 sec |
| Greedy, $\alpha{=}0.1$ | ✓ | $\mathbf{79.1}_{\pm 2.1}$ | 1.2 sec |

Table 4: **Comparing Matching Algorithms** to use for $M_i$ on CIFAR-10 (5+5) joint 10-way accuracy. Permuting B→A as in prior work (Eq. 2) performs poorly. We significantly improve by merging features *within* each model (Eq. 8). Our greedy approach is nearly as accurate as the optimal algorithm while being two orders of magnitude faster.

lated pair of features without replacement. This performs almost as well, and is multiple orders of magnitude faster. If we allow repeated matches (Sec. 4.3), we obtain a slightly better result. Like Bolya et al. (2023), we find that matching is better for merging features than clustering (K-Means).

## 7    CONCLUSION

In this paper, we tackle the extremely difficult task of merging models trained on completely disjoint tasks *without additional training*. We find that prior work underperforms in this setting and posit that they neither fully (1) exploit model similarities nor (2) account for model dissimilarities. We introduce ZipIt!, a general framework for merging models that addresses these issues, and show it to significantly outperform prior work across several difficult settings, comprehensively analyzing each.

**Reproducibility Statement.** To ensure reproducibility, we will release code for our algorithm, experiments, and baselines. We also include algorithm details in Section 4 and further in Appendix C, experimental details in Section 5 and Appendix B, and a proof of our Theorem 1 in Appendix G.

**Acknowledgements.** This work was supported in part by funding from NSF CAREER #2144194, ARL, Google, and NSF GRFP. All views and conclusions expressed in this work are those of the authors and not a reflection of these sources.

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

## A  PARTIAL ZIPPING

In Fig. 7 we plot the average per task accuracy by the number of layers zipped in ResNet-20×8 for CIFAR-100 (50+50) and ResNet-50 for ImageNet-1k (200+200). Note that to avoid adding extra unmerge modules into the network, our stopping point while unzipping has to be the end of a stage.

In Table 5, we show the average neuron correlations at each partial-zipping stage between the layers of ResNet-20 (8×) models trained on the CIFAR-100 (50+50) task. We collect results using the same models used in Table 1b, and compute correlations as described in Section 4. Overall, we find the correlations between models consistently decreases through successive partial zipping locations. This corroborates the finding of Kornblith et al. (2019) that model layers become increasingly dissimilar with depth, as they encode more task-specific features. Coupling Table 5 with Figure 7a, we observe a direct correlation between layer-(dis)similarities and performance decrease. This illustrates the importance of layer similarity between two networks and strong performance.

## B  DATA USAGE

In our approach, we use a sample of the training set in order to compute activations and match features together. For the main paper, we used the full training set for CIFAR, 1% of the training set for ImageNet, and the number of images in the smallest training set for the Multi-Dataset classification experiment (so that we could use the same number of images from each dataset). In each case, we used the same data augmentations from training.

That begs the question: how much data do we actually need, and how necessary are data augmentations? Here we ablate the amount of data used for our CIFAR-100 (50+50) ResNet-20 (8× width) and ImageNet (200+200) Resnet-50 ($22/50$ layers) experiments. In Fig. 8, we test how much data is actually necessary to obtain a good accuracy on CIFAR and ImageNet with or without data augmentation.

We ultimately find that the amount of data doesn't actually matter that much. In the main paper, we use the entire training set for CIFAR-100 with a batch size of 500 (100 batches, or 50,000 images), but it seems like as little as 2 batches (100 images) produces the same result. Similarly on ImageNet, using 0.05% of the data (640 images) produces the same result as 5% (64,048 images).

In fact, the main consideration is whether or not to use data augmentation. For less diverse datasets like CIFAR-100, data augmentation seems essential (giving an almost 4% boost in average task accuracy), and well above the variance of results without augmentation. However, for ImageNet, which has much more diverse images, data augmentation actually hurts slightly on average— though the two are within variance. Note that despite this result, for consistency we use data augmentation in *all* experiments.

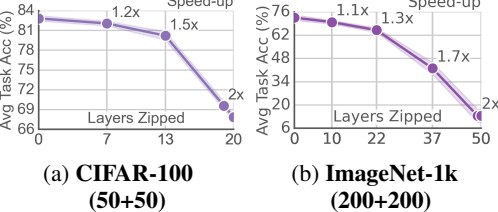

(a) **CIFAR-100 (50+50)**   (b) **ImageNet-1k (200+200)**

Figure 7: **Varying Partial Zip.** By leaving some layers unzipped (Sec. 4.3), we can recover a significant amount of performance while still merging most of the model.

### Average Stage Correlations

| Layer $7/20$ | Layer $13/20$ | Layer $19/20$ |
|---|---|---|
| $0.50_{\pm 0.01}$ | $0.37_{\pm 0.00}$ | $0.27_{\pm 0.00}$ |

Table 5: **CIFAR-100 (50+50) Zipping Correlations.** We show the average correlations between two ResNet-20 (8× width) models at each partial zipping stage. Correlations consistently decrease at each successive stage, indicating that the layers of the two models increasingly diverge.

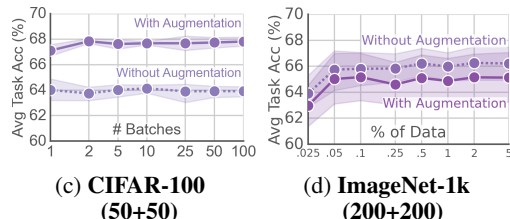

(c) **CIFAR-100 (50+50)**   (d) **ImageNet-1k (200+200)**

Figure 8: **Data Usage.** How much data do we need to use to compute activations? We find that only a few hundred images are needed to obtain the best performance. Data augmentation is not always useful.

## C    ZIP PROPAGATION DETAILS

In the main paper we described general rules for zip propagation—namely, propagate through layers until you reach a module with a weight matrix. Here, we describe rules more concretely for each layer type needed to define most convnets.

**Linear.** Apply $M_i$ and $U_i$. Stop propagation.

**Conv.** Apply $M_i$ and $U_i$ to each kernel location (i.e., move the $k \times k$ kernel dimensions to the batch dimension). Stop propagation.

**BatchNorm.** Apply $M_i$ to all parameters (weight, bias, mean, variance), squaring it for the variance term. Continue propagation. As Jordan et al. (2022) points out, we cannot compute the correct variance without knowing the covariance between the two models (which we don't have access to). Thus, we reset batch norms after merging to evaluate the variance correctly.

**LayerNorm.** Apply $M_i$ to all parameters (weight, bias). Continue propagation. Since LayerNorm computes mean and standard deviation on the fly, we don't need to do anything special.

**ReLU.** Nothing to merge. Continue propagation. Note that passing the merge matrix unchanged through the ReLU layers is an approximation, since we're using a linear merge operation on nonlinear features. Addressing this issue could be an interesting topic for future work, as even the permute and add approach of prior work has this issue (ReLU is invariant to permutation, but certainly not adding).

**Avg / Max Pool.** Nothing to Merge. Continue propagation.

**Skip Connection.** Continue propagation through every input to the skip connection (using the same $M_i$ and $U_i$ for each).

## D    CROSS ENTROPY ON CIFAR

In the main paper, we train our CIFAR models with a CLIP (Radford et al., 2021) loss (using CLIP embeddings of class names as targets). This ensures that the output spaces of the two models are aligned, which is necessary to get good accuracy for prior work that merge the entire model together.

| Method | FLOPs (G) | Joint | Task A | Task B | Avg |
|---|---|---|---|---|---|
| Model A | 10.88 | 37.9 | 74.15 | 1.7 | 36.4 |
| Model B | 10.88 | 36.7 | 2.2 | 75.2 | 38.7 |
| W. Avg | 10.88 | 2.7 | 5.0 | 4.9 | 4.9 |
| Git Re-Basin‡ | 10.88 | 3.1 | 5.8 | 5.3 | 5.6 |
| Permute | 10.88 | 20.0 | 30.8 | 32.8 | 31.8 |
| **ZipIt!**$_{20/20}$ | 10.88 | **27.9** | **40.1** | **39.7** | **39.9** |
| Ensemble | 21.76 | 60.5 | 74.2 | 75.2 | 74.7 |
| **ZipIt!**$_{13/20}$ | 14.52 | 38.6 | 51.8 | 52.0 | 51.9 |
| **ZipIt!**$_{7/20}$ | 18.14 | **47.0** | **60.6** | **60.5** | **60.6** |

Table 6: **CIFAR-100 (50+50) Cross Entropy.** ZipIt! vs. baselines using ResNet-20 ($16\times$ width). Merging the entire model as in prior work produces bad results when using cross-entropy, hence we use CLIP in the main draft. If we use partial zipping, we can recover a lot of the lost performance. ‡ refers to (Ainsworth et al., 2022)

**ResNet.** In Tab. 6, we show results for CIFAR-100 (50+50) where we train with the normal one-hot targets (i.e., like we did for ImageNet), instead. Immediately, we see that accuracies of the merged models are much lower across the board, with no method able to outperform just using one of the original models when merging the entire network. In fact, Git Re-Basin (Ainsworth et al., 2022) does almost no better than weight averaging, which gets close to random accuracy. While ZipIt! without partial zipping also performs worse than the original models, it still greatly outperforms all prior work. And with partial zipping, ZipIt! is able to exceed the accuracy of the original models.

Thus, in the case of using cross-entropy loss, partial zipping is extremely important. Merging the entire model as in prior work fails, since the later layers of the model are incompatible with each other due to each model having a different output space. Partial zipping, on the other hand, can mitigate that issue.

**VGG.** In the main paper, we use ResNets for each experiment, since they are easy to train and produce strong results. However, in principle ZipIt! can work on any architecture. For completeness, in Tab. 7, we show results on the CIFAR-10 (5+5) setting with VGG11 (1× width). Note that this is a much smaller and weaker model than the ResNet-20s we use in the main paper, so its results on CIFAR-10 aren't as strong. Furthermore, we conducted this experiment with a cross entropy loss, so merging the entire model performs worse than the original models.

|  |  | Accuracies (%) | | | |
|---|---|---|---|---|---|
| Method | FLOPs (G) | Joint | Task A | Task B | Avg |
| Model A | 0.15 | 44.6 | 89.2 | 21.0 | 55.1 |
| Model B | 0.15 | 44.0 | 23.1 | 88.1 | 55.6 |
| W. Avg | 0.15 | 10.2 | 20.8 | 20.9 | 20.9 |
| Permute | 0.15 | 25.4 | 47.2 | 48.5 | 47.8 |
| **ZipIt!**$_{22/22}$ | 0.15 | **33.2** | **53.8** | **59.9** | **56.5** |
| Ensemble | 0.30 | 66.6 | 89.2 | 88.1 | 88.6 |
| **ZipIt!**$_{14/22}$ | 0.17 | 35.2 | 56.7 | 60.2 | 58.4 |
| **ZipIt!**$_{7/22}$ | 0.27 | **44.5** | **66.0** | **65.1** | **65.5** |

Table 7: **CIFAR-10 (5+5) CE with VGG.** ZipIt! vs. baselines using VGG11 (1× width) using Cross Entropy instead of CLIP loss. ZipIt! displays the same behavior here as it does for ResNet-20 with low width.

Despite this, we observe a very similar trend to the ResNet-20 models in that ZipIt! outperforms all baselines and that partial zipping is important for reaching the accuracy of the original models (in this case, matching not exceeding). In fact, these results continue a more general trend in that ZipIt! greatly benefits from larger model scales, making effective use of the extra capacity. In this case, the scale of the model is quite small, so there is not as much room in the weights to store the potentially disjoint features of both models.

# E    IMAGENET WITH 1.5X WIDTH

In the main paper, we show that ZipIt! scales very well with increased width of the model for the CIFAR-100 (50+50) setting. While CIFAR-100 is a challenging dataset on its own, the natural question is if that same trend occurs the much harder ImageNet-1k (200+200) setting.

In Tab. 8, we test this by comparing ZipIt! on the original 1× width ResNet-50 in the main paper with a 1.5× width one. In all cases, except for the fully zipped model (likely because of the Cross-Entropy loss), ZipIt! enjoys a large jump in performance from the extra width. For 37 layers, 33.1% joint accuracy becomes 49.0%. For 22 layers, 55.8% becomes 64.1%. And for 10 layers, 60.9% becomes 66.8%, now only 1% away from the ensemble. Thus, even in this much more challenging setting, ZipIt! is able to make full use of the extra model capacity.

|  |  | Accuracies (%) | | | |
|---|---|---|---|---|---|
| Method | FLOPs (G) | Joint | Task A | Task B | Avg |
| | | 1× Width | | | |
| Ensemble | 8.22 | 63.3 | 74.3 | 70.5 | 72.4 |
| **ZipIt!**$_{50/50}$ | 4.11 | 8.6 | 12.4 | 14.7 | 13.5 |
| **ZipIt!**$_{37/50}$ | 4.92 | 33.1 | 41.8 | 42.3 | 42.0 |
| **ZipIt!**$_{22/50}$ | 6.39 | 55.8 | 65.9 | 64.1 | 65.0 |
| **ZipIt!**$_{10/50}$ | 7.43 | 60.9 | 70.7 | 69.0 | 69.9 |
| | | 1.5× Width | | | |
| Ensemble | 32.6 | 67.8 | 76.7 | 72.6 | 74.7 |
| **ZipIt!**$_{50/50}$ | 16.3 | 9.7 | 13.2 | 16.0 | 14.6 |
| **ZipIt!**$_{37/50}$ | 19.5 | 49.0 | 56.2 | 56.7 | 56.4 |
| **ZipIt!**$_{22/50}$ | 25.5 | 64.1 | 71.6 | 70.4 | 71.0 |
| **ZipIt!**$_{10/50}$ | 29.7 | 66.8 | 74.9 | 72.1 | 73.5 |

Table 8: **ImageNet-1k (200+200) Width Comparison.** We show how ZipIt! is able to make use of the extra model width when merging models together. For instance, merging 37 layers goes from 33% joint accuracy with 1× width to 49% with 1.5×, while the ensemble only improves by 4%. These models use cross-entropy, so merging the entire network results in poor performance.

# F    MERGING MODELS WITH DIFFERENT OUTPUT MODALITIES

In this experiment we use ZipIt! to merge two models with different initializations trained on different tasks with *different* output modalities: semantic segmentation and image classification. Specifically, we merge the ResNet-50 backbone of a DeepLabV3 (Chen et al., 2017) model finetuned on the Pascal VOC (Everingham et al., 2010) dataset, with a ResNet-50 model trained on ImageNet-1k. While the DeepLabV3 backbone was itself pre-trained on ImageNet-1k, it was further finetuned on Pascal VOC and does not share the initialization of our classification model. Table 9 shows the results of combining the two ResNet-50 models with ZipIt! at various partial

| Method | Accuracy (%) ImageNet-1k | mIoU (%) Pascal VOC |
|---|---|---|
| W. Avg | 0.8 | 3.3 |
| **ZipIt!**$_{49/50}$ | **23.1** | **6.0** |
| Ensemble | 77.8 | 76.8 |
| **ZipIt!**$_{37/50}$ | 47.7 | 35.0 |
| **ZipIt!**$_{22/50}$ | 60.9 | 64.4 |
| **ZipIt!**$_{10/50}$ | **64.9** | **71.7** |

Table 9: **PASCAL VOC and ImageNet-1k** merging models with different output modalities using a DeepLabV3 ResNet-50 backbone and ImageNet-1k Resnet-50 model.

merging locations. We evaluate the performance of each merged by reporting its ImageNet-1k accuracy, and its Pascal VOC mIoU as is standard. Overall, we observe that ZipIt! is capable of merging nearly half the number of ResNet-50 layers between both models while still maintaining good performance on *both tasks*, all *without any training*.

# G  A TIGHTER BOUND FOR LINEAR MODE CONNECTIVITY

In this section we demonstrate that merging models by supporting feature merges both *across* and *within* each, yields a tighter bound than Theorem 3.1 in (Entezari et al., 2021) in its limited setting. We first introduce necessary background from prior work, including Theorem 3.1 and a particular formalization for within-model merging borrowed from (Simsek et al., 2021). Second, we introduce Theorem 1, which produces a tighter bound on Theorem 3.1 when merging within models is allowed, and prove its validity (Section G.2 & G.3). Third, we provably extend Theorem 1 to a less restrictive setting, retaining its bounds (Section G.4).

## G.1  BACKGROUND

We first introduce Theorem 3.1 from (Entezari et al., 2021). Second, we formalize a restricted version of within-model merging necessary for our proof using the definitions from (Simsek et al., 2021).

### G.1.1  THOEREM 3.1

**The Theorem.**  Let $f_{\{\mathbf{v}, \mathbf{W}\}}(x) = \mathbf{v}^T \sigma(\mathbf{W}x)$, $f_{\{\mathbf{v}', \mathbf{W}'\}}(x) = \mathbf{v}'^T \sigma(\mathbf{W}'x)$ be two fully-connected networks with $h$ hidden units where $\sigma(\cdot)$ is ReLU activation, $\mathbf{v} \in \mathbb{R}^h$ and $\mathbf{W} \in \mathbb{R}^{h \times d}$ are the parameters and $x \in \mathbb{R}^d$ is the input. If each element of $\mathbf{W}$ and $\mathbf{W}'$ is sampled uniformly from $[-1/\sqrt{d}, 1/\sqrt{d}]$ and each element of $\mathbf{v}$ and $\mathbf{v}'$ is sampled uniformly from $[-1/\sqrt{h}, 1/\sqrt{h}]$, then for any $x \in \mathbb{R}^d$ such that $\|x\|_2 = \sqrt{d}$, with probability $1 - \delta$ over $\mathbf{W}, \mathbf{W}', \mathbf{v}, \mathbf{v}'$, there exist a permutation such that

$$|f_{\{\alpha\mathbf{v}+(1-\alpha)\mathbf{v}'', \alpha\mathbf{W}+(1-\alpha)\mathbf{W}''\}}(x) - \alpha f_{\{\mathbf{v}, \mathbf{W}\}}(x) - (1-\alpha)f_{\{\mathbf{v}', \mathbf{W}'\}}(x)| = \tilde{O}(h^{-\frac{1}{2d+4}}) \quad (9)$$

where $\mathbf{v}'', \mathbf{W}''$ are permuted version of $\mathbf{v}', \mathbf{W}'$, $\alpha \in [0, 1]$ is an arbitrary interpolation constant, and the left-hand-side of the equality is the amount an interpolated model differs in output compared to the interpolation of the original models. (Entezari et al., 2021) show that minimizing this quantity is analogous to minimizing the barrier (as defined by Entezari et al. (2021)) in this setting. This is important because it states that achieving a zero output difference is equivalent to achieving zero-barrier, which implies that two models are linearly mode connected (LMC).

**Implications** Theorem 3.1 states that given any two two-layer models with different random initializations, there exists a permutation for one model such that applying the permutation makes it linearly mode connected to the second with high probability, given that the networks are *wide enough* (i.e. $h$ is large enough). In other words, it states that any two randomly initialized two-layer networks are LMC modulo permutation with high likelihood. Entezari et al. (2021) use this result to conjecture that most well-trained neural networks with the same architecture and trained on the same task are also LMC modulo permutation with high likelihood.

Notably however, permutations only allow for merging *across* models. We will show how adding the ability to merge *within* models leads to a tighter bound than Theorem 3.1 with the same likelihood.

### G.1.2  A RESTRICTED FORMALIZATION OF MERGING WITHIN MODELS

**The Formalization.**  Let $\boldsymbol{\theta}_h = \{\mathbf{v}, \mathbf{W}\}$ represent a parameter-set such that $f_{\{\mathbf{v}, \mathbf{W}\}} = f_{\boldsymbol{\theta}_h}$, and likewise let $\boldsymbol{\theta}'_h = \{\mathbf{v}', \mathbf{W}'\}$, s.t. , $f_{\{\mathbf{v}', \mathbf{W}'\}} = f_{\boldsymbol{\theta}'_h}$. Given $\boldsymbol{\theta}_h$, let $\Theta_h$ denote the set of all parameter-sets with functional equivalence to $\boldsymbol{\theta}_h$. This means that $\forall \theta \in \Theta_h$, and $\forall x \in \{x \in \mathbb{R}^d | \|x\|_2 = \sqrt{d}\}$, $f_{\boldsymbol{\theta}}(x) = f_{\boldsymbol{\theta}_h}(x)$. Similarly, let $\Theta'_h$ be the set of all parameter-sets with functional equivalence to $\boldsymbol{\theta}'_h$. Following $\boldsymbol{\theta_h}$, let $\boldsymbol{\theta}_r$ be an arbitrary parameter-set for $f$ which has $r$ hidden units instead. Assume $\boldsymbol{\theta}_h$ can be reduced to some $\boldsymbol{\theta}_r, r \leq h$ in a function-invariant manner using the definition of

zero-type neurons from (Simsek et al., 2021). This means that there are $h - r$ total combinations of (1) rows in $\mathbf{W}$ that are copies of one another whose corresponding $\mathbf{v}$ elements *sum to* 0, and (2) some zero-elements in $\mathbf{v}$. Thus, following Simsek et al. (2021) there exists a function-and-loss-preserving affine transformation that reduces $\boldsymbol{\theta}_h$ to $\boldsymbol{\theta}_r$. We denote this function as $M_{h \to r} \in \mathbb{R}^{r \times h}$, with $M_{h \to r}(\boldsymbol{\theta}_h) = \boldsymbol{\theta}_r$. Note that when $r = h$, $M_{h \to r}$ can simply be the identity transformation.

By definition, $\boldsymbol{\theta}_h$ lies in the expansion manifold of $\boldsymbol{\theta}_r$ (Simsek et al., 2021). This means there is a similarly defined affine transformation $U_{r \to h} \in \mathbb{R}^{h \times r}$ that can expand $\boldsymbol{\theta}_r$ back to arbitrary $\tilde{\boldsymbol{\theta}}_{\boldsymbol{h}} \in \Theta_h$ lying on the expansion manifold. One simple way is to extend $\boldsymbol{\theta}_r$ to $\tilde{\boldsymbol{\theta}}_{\boldsymbol{h}}$ by filling the remaining $h - r$ $\mathbf{v}$ elements with 0 and the $h - r$ $\mathbf{W}$ rows with arbitrary values. Because the associated $\mathbf{v}$ elements for each $\mathbf{W}$ row are zero, the values of each row don't impact the function output. Note that because $h \geq r$, $U_{r \to h}$ can assign $\boldsymbol{\theta}_r$ into arbitrary new indices in $\boldsymbol{\theta}_h$. Thus, $U_{r \to h}$ act as both a *permutation and expansion* transformation. Let $T = U \circ M = U_{r \to h}(M_{h \to r}(\boldsymbol{\theta}_h))$ be the coupling of the reduction and expansion affine-transformations that produce new networks of width $h$ from $\boldsymbol{\theta}_r$. By definition, any $T$ is a permutation when $M$ is the identity and $U$ is the permutation matrix. For the remainder of this section, we assume that $T$ further contains a permutation (i.e. $T = P \circ U \circ M$ for some permutation matrix $P \in \mathbb{R}^{h \times h}$).

We will leverage the concept of zero-type neurons presented in (Simsek et al., 2021) to obtain a tighter bound on Theorem 3.1.

**A Note on Novelty.** While we borrow ideas from Simsek et al. (2021), our Theorem is a differs in theoretical application. First, Simsek et al. (2021) restrict their attention to overall connectivity across points within expansion manifolds. This is important because our Theorem and proof do not require models to lie on the *same* expansion manifold to be linearly mode connected. Second, our models need not be reducible to the same $r$. That is, we allow for arbitrary reducibility between any two network parameter-sets. Our theorem also differs from Theorem 3.1 in (Entezari et al., 2021) in that we extend function-invariant transformations beyond the permutation matrix, and show that tighter bounds are achievable in the process. Furthermore, we show that uniformity assumptions may be relaxed while retaining the same bounds (Section G.4).

## G.2   A THEORETICAL RESULT

We now introduce Theorem 1, an extension of Theoerem 3.1 that yields a strictly tighter bound when the transformations $T$ from Section G.1.2 are included and $r < h$, and exactly as tight when $r = h$. We leave the proof to the next section.

**Theorem 1.** Let $f_{\{\mathbf{v},\mathbf{W}\}}(x) = \mathbf{v}^T \sigma(\mathbf{W}x)$, $f_{\{\mathbf{v}',\mathbf{W}'\}}(x) = \mathbf{v}'^T \sigma(\mathbf{W}'x)$ be two fully-connected networks with $h$ hidden units where $\sigma(\cdot)$ is ReLU activation, $\mathbf{v} \in \mathbb{R}^h$ and $\mathbf{W} \in \mathbb{R}^{h \times d}$ are the parameters and $x \in \mathbb{R}^d$ is the input. If each element of $\mathbf{W}$ and $\mathbf{W}'$ is sampled uniformly from $[-1/\sqrt{d}, 1/\sqrt{d}]$ and each element of $\mathbf{v}$ and $\mathbf{v}'$ is sampled uniformly from $[-1/\sqrt{h}, 1/\sqrt{h}]$, then for any $x \in \mathbb{R}^d$ such that $\|x\|_2 = \sqrt{d}$, with probability $1 - \delta$ over $\mathbf{W}, \mathbf{W}', \mathbf{v}, \mathbf{v}'$, there exist transformations $T, T'$ such that

$$|f_{\{\alpha\tilde{\mathbf{v}}+(1-\alpha)\tilde{\mathbf{v}}', \alpha\tilde{\mathbf{W}}+(1-\alpha)\tilde{\mathbf{W}}'\}}(x) - \alpha f_{\{\mathbf{v},\mathbf{W}\}}(x) - (1-\alpha)f_{\{\mathbf{v}',\mathbf{W}'\}}(x)|$$
$$\leq \begin{cases} \tilde{O}\left( \left(\frac{h^2}{(r+r')-h}\right)^{-\frac{1}{2d+4}} \right) & , (r+r') - h > 0 \\ 0 & , \text{otherwise} \end{cases} \tag{10}$$

where $\tilde{\mathbf{v}}, \tilde{\mathbf{W}}$ are transformed versions of $\mathbf{v}, \mathbf{W}$ from $T$ and $\tilde{\mathbf{v}}', \tilde{\mathbf{W}}'$ are transformed versions of $\mathbf{v}', \mathbf{W}'$ from $T'$ respectively. $0 < r, r' \leq h$ are the hidden unit amounts each network can be reduced to via its respective $M, M'$ transformation before being expanded back to width $h$ via $P \circ U, P' \circ U'$, where $P, P'$ are permutation matrices.

**Implications.** Theorem 1 states that when redundancy exists and can be leveraged in a network, one can find a transformation that yields strictly lower barrier than with permutation with any $h$. Moreover, it approaches zero-barrier faster with increase in $h$ compared to permutations. Although it only explicitly holds for random initializations—like Theorem 3.1, this theoretical intuition is supported by our experimental observations. For instance it explains why algorithms like ZipIt!

appear to converge to the ensemble exponentially faster than permutation methods in Figure 6b). The ensemble achieves zero-barrier, and ZipIt! is faster to approach it than permutation counterparts because it can reduce models with minimal deduction in performance.

## G.3  THEOREM 1 PROOF

We now derive our proposed Theorem 1. Theorem 1 is very similar to Theorem 3.1— we just add the reducibility property from Section G.1.2. Thus, our derivation is nearly identical to their Appendix D proof. We fully derive the novel components of Theorem 1 for clarity, while referring to Appendix D in (Entezari et al., 2021) for the remaining identical derivations to avoid redundancy.

Let $\boldsymbol{\theta}_h = \{\mathbf{v}, \mathbf{W}\}$ and $\boldsymbol{\theta}'_h = \{\mathbf{v}', \mathbf{W}'\}$ respectively as defined in Section G.1. Suppose each can be reduced to some $\boldsymbol{\theta}_r, \boldsymbol{\theta}'_r$ respectively with $r \leq h$, via an appropriate $M_{h \to r}, M_{h \to r'}$ transformation. Further, let $U_{r \to h}, U_{r' \to h}$ be as defined in Section G.1, expanding $M_{h \to r}(\boldsymbol{\theta}_h), M_{h \to r'}(\boldsymbol{\theta}_h)$ back to width $h$ by filling in all $h - r, h - r'$ dimensions in $\mathbf{v}, \mathbf{v}'$ with 0 respectively and all $h - r, h - r'$ dimensions in $\mathbf{W}, \mathbf{W}'$ with some specific values. Finally, let $T, T'$ be transformations defined in Section G.1 with $T = P \circ U \circ M, T' = P' \circ U' \circ M'$ respectively.

Let $\tilde{\boldsymbol{\theta}}_h = T(\boldsymbol{\theta}_h), \tilde{\boldsymbol{\theta}}'_h = T'(\boldsymbol{\theta}'_h)$ be the new parameter-sets obtained from $T, T'$ respectively, and let $\tilde{\boldsymbol{\theta}}_h = \{\tilde{\mathbf{v}}, \tilde{\mathbf{W}}\}$ and $\tilde{\boldsymbol{\theta}}'_h = \{\tilde{\mathbf{v}}', \tilde{\mathbf{W}}'\}$. By definition, $\tilde{\boldsymbol{\theta}}_h \in \Theta_h$, and $\tilde{\boldsymbol{\theta}}'_h \in \Theta'_h$. From the definitions of $T, T', \mathbf{W}$ has $h - r$ zero $\tilde{\mathbf{v}}$ elements and $\mathbf{W}'$ has $h - r'$ zero-$\tilde{\mathbf{v}}'$ elements. Now, let us suppose that the corresponding $h - r$ rows in $\tilde{\mathbf{W}}$ are set to copy rows in $\tilde{\mathbf{W}}'$, and similarly $h - r'$ rows in $\tilde{\mathbf{W}}'$ rows are set to copy rows in $\tilde{\mathbf{W}}$. Now, interpolating between any non-zero element and a zero-element is equivalent to simply scaling the non-zero element: $z : \alpha 0 + (1 - \alpha)z = (1 - \alpha)z$. Thus, so long as $h \leq (h - r) + (h - r')$, we can achieve *perfect* interpolation by placing $h - r$ elements from $\tilde{\boldsymbol{\theta}}'_h$ into the zero-elements of $\tilde{\boldsymbol{\theta}}_h$ and $h - (h - r) \leq h - r'$ elements from $\tilde{\boldsymbol{\theta}}_h$ into the zero-elements of $\tilde{\boldsymbol{\theta}}'_h$. This yields the zero-part of our piece-wise bound. However, the proof is more involved for the second case when $h > (h - r) + (h - r') \to (r + r') - h > 0$. We continue the proof for this case below.

First, note that we only need to worry about the $(r + r') - h$ rows in $\mathbf{W}, \mathbf{W}'$ that cannot necessarily be matched with perfect interpolation as shown above. Let $\mathbb{K}$, and $\mathbb{K}'$ be the set of these rows for each network respectively, where $|\mathbb{K}| = |\mathbb{K}'| = (r + r') - h$. These are the only rows within the two models that must still be considered. For any given $\xi > 0$, we consider the set $S_\xi = \{-1/\sqrt{d} + \xi, -1/\sqrt{d} + 3\xi, \ldots, 1/\sqrt{d} - \xi\}^d$, a discretization of the $\mathbb{R}^d$ which has size $(\frac{1}{\xi\sqrt{d}})^d$[1]. For any $s \in S_\xi$, let $C_s(\tilde{\mathbf{W}})$ be the set of indices of rows in $\mathbb{K}$ of $\tilde{\mathbf{W}}$ that are closest in Euclidean distance to $s$ than any other element in $S_\xi$:

$$C_s(\tilde{\mathbf{W}}) = \{i | \boldsymbol{w}_i \in \mathbb{K}, s = \arg\min_{s' \in S_\xi} \|\mathbf{w}_i - s'\|_\infty\}$$
$$C_s(\tilde{\mathbf{W}}') = \{i | \boldsymbol{w}_i \in \mathbb{K}', s = \arg\min_{s' \in S_\xi} \|\mathbf{w}_i - s'\|_\infty\}$$

where for simplicity we assume that arg min returns a single element. These are the same definitions and assumptions as in (Entezari et al., 2021).

Now for every $s \in \mathcal{S}$ consider a random 1-1 matching (permutation) of elements in $C_s(\tilde{\mathbf{W}})$ and $C_s(\tilde{\mathbf{W}}')$. Whenever $|C_s(\tilde{\mathbf{W}})| \neq |C_s(\tilde{\mathbf{W}}')|$, we will inevitably have unmatched indices because permutations only allow 1-1 mappings. Let $I$, and $I'$ denote the set of total unmatched indices from $\mathbb{K}$, and $\mathbb{K}'$ respectively. If $|C_s(\tilde{\mathbf{W}})| - |C_s(\tilde{\mathbf{W}}')| \geq 0$, we add these extra indices that are not matched to $I$ and otherwise we add them to $I'$. Because $\mathbb{K}$, and $\mathbb{K}'$ are the same size, $|I| = |I'|$ after adding all unmatched indices across all $C_s$. Thus, by definition $|I| = |I'| \leq (r + r') - h$.

Pick an arbitrary $s \in \mathcal{S}$. Since each element of $\mathbb{K}$ and $\mathbb{K}'$ is sampled uniformly from $[-1/\sqrt{d}, 1/\sqrt{d}]$, for each row in the respective sets, the probability of being assigned to each $s \in \mathcal{S}_\xi$ is a multinomial distribution with equal probability for each $s$: $1/|\mathcal{S}_\xi|$. Pick an arbitrary $s$. $|C_s(\tilde{\mathbf{W}})|$ is the sum over all indicator variables $W_i^s = \mathbb{1}\{w_i \in C_s(\tilde{\mathbf{W}})\}$, and the expected value of this sum, $E[|C_s(\tilde{\mathbf{W}})|] = [(r + r') - h]/|\mathcal{S}_\xi|$ as there are $(r + r') - h$ total rows. Let $(r + r') - h = n$. Since each $W_i^s$ is between

---

[1]Like (Entezari et al., 2021), we choose $\xi$ such that it is a factor of $1/\sqrt{d}$.

$[0, 1]$, we can use Hoeffding's Inequality to bound the size of $C_s(\tilde{\mathbf{W}})$ with high probability

$$P(|S_n - E[S_n]| \geq t) \leq 2 \exp\left(\frac{-2t^2}{n}\right) \quad \text{(Hoeffding's Inequality)} \quad (11)$$

$$P(||C_s(\tilde{\mathbf{W}})| - E[|C_s(\tilde{\mathbf{W}})|]| \geq t) \leq 2 \exp\left(\frac{-2t^2}{n}\right) \qquad \because S_n = |C_s(\tilde{\mathbf{W}})| \quad (12)$$

$$P\left(||C_s(\tilde{\mathbf{W}})| - \frac{(r+r')-h}{|\mathcal{S}_\xi|}| \geq t\right) \leq 2 \exp\left(\frac{-2t^2}{n}\right) \quad (13)$$

$$P\left(||C_s(\tilde{\mathbf{W}})| - \frac{(r+r')-h}{|\mathcal{S}_\xi|}| \geq t\right) \leq 2 \exp\left(\frac{-2t^2}{(r+r')-h}\right) \qquad \because n = (r+r') - h \quad (14)$$

Let $n = (r + r') - h$. By taking a union bound over all elements of $\mathbb{K}$ and $\mathbb{K}'$, with probability $1 - \delta/3$, the following holds for all choices of $s$:

$$\frac{n}{|S_\xi|} - \sqrt{\frac{n}{2} \log(12|S_\xi|/\delta)} \leq |C_s(\tilde{\mathbf{W}})|, |C_s(\tilde{\mathbf{W}}')| \leq \frac{n}{|S_\xi|} + \sqrt{\frac{n}{2} \log(12|S_\xi|/\delta)} \quad (15)$$

Note this derivation almost exactly follows (Entezari et al., 2021), except that we have $n \leq h$, yielding a tighter size-bound.

Using Eq. (15), we can obtain a bound on the cardinality differences between $C_s(\tilde{\mathbf{W}})$ and $C_s(\tilde{\mathbf{W}}')$ by subtracting the minimum value of one from the maximum value of the other:

$$||C_s(\tilde{\mathbf{W}})| - |C_s(\tilde{\mathbf{W}}')|| \leq \sup(|C_s(\tilde{\mathbf{W}})|) - \inf(|C_s(\tilde{\mathbf{W}})|) \quad (16)$$

$$||C_s(\tilde{\mathbf{W}})| - |C_s(\tilde{\mathbf{W}}')|| \leq \left(\frac{n}{|S_\xi|} + \sqrt{\frac{n}{2} \log(12|S_\xi|/\delta)}\right)$$
$$- \left(\frac{n}{|S_\xi|} - \sqrt{\frac{n}{2} \log(12|S_\xi|/\delta)}\right) \quad (17)$$

$$||C_s(\tilde{\mathbf{W}})| - |C_s(\tilde{\mathbf{W}}')|| \leq 2\sqrt{\frac{n}{2} \log(12|S_\xi|/\delta)} \quad (18)$$

Using Eq. (18) we can bound the size of $I, I'$ with probability $1 - \delta/3$ as follows:

$$\sum_{s \in S_\xi} ||C_s(\tilde{\mathbf{W}})| - |C_s(\tilde{\mathbf{W}}')| \leq \sum_{s \in S_\xi} 2\sqrt{\frac{n}{2} \log(12|S_\xi|/\delta)} \quad (19)$$

$$\sum_{s \in S_\xi} ||C_s(\tilde{\mathbf{W}})| - |C_s(\tilde{\mathbf{W}}')| \leq 2|S_\xi|\sqrt{\frac{n}{2} \log(12|S_\xi|/\delta)} \quad (20)$$

$$|I| + |I'| = \sum_{s \in S_\xi} ||C_s(\tilde{\mathbf{W}})| - |C_s(\tilde{\mathbf{W}}')| \leq 2|S_\xi|\sqrt{\frac{n}{2} \log(12|S_\xi|/\delta)} \quad (21)$$

$$|I| = |I'| = \frac{1}{2}\sum_{s \in S_\xi} ||C_s(\tilde{\mathbf{W}})| - |C_s(\tilde{\mathbf{W}}')| \leq |S_\xi|\sqrt{\frac{n}{2} \log(12|S_\xi|/\delta)} \qquad \because |I| = |I'| \quad (22)$$

Note, our Eq. (22) equivalent to Eq. (6) in (Entezari et al., 2021), but in terms of $n$ instead of $h$.

The remainder of our derivation exactly follows (and achieves identical bounds to) (Entezari et al., 2021) until directly after their substitution of $\xi$. To avoid writing an identical derivation, we refer readers to their derivation following Eq. (6) in their Appendix D until the substitution of $\xi$, and instead pick up immediately before the substitution of $\xi$:

Let $\epsilon \geq 0$ denote the value of $|f_{\alpha\tilde{\mathbf{v}}+(1-\alpha)\tilde{\mathbf{v}}', \alpha\mathbf{W}+(1-\alpha)\tilde{\mathbf{W}}'}(x) - \alpha f_{\{\mathbf{v},\mathbf{W}\}}(x) - (1-\alpha)f_{\{\mathbf{v}',\mathbf{W}'\}}(x)|$. Following the derivation of (Entezari et al., 2021), we bound $\epsilon$ as follows:

$$\epsilon = |f_{\{\alpha\tilde{\mathbf{v}}+(1-\alpha)\tilde{\mathbf{v}}'\}, \{\alpha\mathbf{W}+(1-\alpha)\tilde{\mathbf{W}}'\}}(x) - \alpha f_{\{\mathbf{v},\mathbf{W}\}}(x) - (1-\alpha)f_{\{\mathbf{v}',\mathbf{W}'\}}(x)|$$
$$\leq \sqrt{2\log(12/\delta)\log(12h/\delta)\left(\frac{|I|}{h} + \xi^2 d\right)} \quad (23)$$

Setting $\xi = \epsilon / \sqrt{4d \log(12/\delta) \log(12h/\delta)}$ gives the following bound on $h$:

$$h \leq \frac{4 \log(12/\delta) \log(12h/\delta)|I|}{\epsilon^2}$$

$$\leq \frac{4 \log(12/\delta) \log(12h/\delta)|S_\xi| \sqrt{\frac{n}{2} \log(12|S_\xi|/\delta)}}{\epsilon^2}$$

Therefore, we have:

$$h^2 \leq \left( \frac{4 \log(12/\delta) \log(12h/\delta)|S_\xi| \sqrt{\frac{n}{2} \log(12|S_\xi|/\delta)}}{\epsilon^2} \right)^2 \tag{24}$$

$$\leq \left( \frac{4 \log(12/\delta) \log(12h/\delta)|S_\xi| \sqrt{\log(12|S_\xi|/\delta)}}{\epsilon^2} \right)^2 \left( \frac{n}{2} \right) \qquad \because |S_\xi)| = \left( \frac{1}{\xi \sqrt{d}} \right)^d \tag{25}$$

$$\leq \left( \frac{4 \log(12/\delta) \log(12h/\delta)}{\epsilon^2} \right)^{d+2} (\log(12/\delta) + d \log(1/\epsilon))(n) \tag{26}$$

$$\frac{h^2}{n} \leq \left( \frac{4 \log(12/\delta) \log(12h/\delta)}{\epsilon^2} \right)^{d+2} (\log(12/\delta) + d \log(1/\epsilon)) \tag{27}$$

Using the inequality in equation (27), we have $\epsilon = \tilde{O}((\frac{h^2}{n})^{-\frac{1}{2d+4}}) = \tilde{O}((\frac{h^2}{r+r'-h})^{-\frac{1}{2d+4}})$.

Thus, we obtain the following piece-wise bound over the barrier:

$$|f_{\{\alpha \tilde{\mathbf{v}} + (1-\alpha)\tilde{\mathbf{v}}', \alpha \bar{\mathbf{W}} + (1-\alpha)\bar{\mathbf{W}}'\}}(x) - \alpha f_{\{\tilde{\mathbf{v}}, \bar{\mathbf{W}}\}}(x) - (1-\alpha) f_{\{\tilde{\mathbf{v}}', \bar{\mathbf{W}}'\}}(x)|$$

$$\leq \begin{cases} \tilde{O} \left( \left( \frac{h^2}{(r+r')-h} \right)^{-\frac{1}{2d+4}} \right) & , (r+r') - h > 0 \\ 0 & , \text{otherwise} \end{cases}$$

$\square$

### G.4 UNIFORMITY IS NOT NEEDED: AN EXTENSION OF THEOREM 1

Although Theorem 1 demonstrates a tighter bound compared to Theorem 3.1 is possible when merging within a model is allowed, its reliance on $\mathbf{W}, \mathbf{v}, \mathbf{W}', \mathbf{v}'$ being uniform random variables is unnecessary. Instead, we can assume that $\mathbf{W}, \mathbf{W}'$ are sampled from an arbitrary probability distribution that is bounded on $[-1/\sqrt{d}, 1/\sqrt{d}]$. Similarly, assume that $\mathbf{v}, \mathbf{v}'$ is sampled from an arbitrary probability distribution that is both centered and bounded on $[-1/\sqrt{h}, 1/\sqrt{h}]$. Note how for both $\mathbf{v}$, and $\mathbf{W}$ *any continuous probability distribution is valid*, so long as it satisfies the stated conditions. We formalize this as follows:

**Theorem 1.1.** Let $f_{\mathbf{v}, \mathbf{W}}(x) = \mathbf{v}^T \sigma(\mathbf{W}x)$, $f_{\mathbf{v}', \mathbf{W}'}(x) = \mathbf{v}'^T \sigma(\mathbf{W}'x)$ be two fully-connected networks with $h$ hidden units where $\sigma(\cdot)$ is ReLU activation, $\mathbf{v} \in \mathbb{R}^h$ and $\mathbf{W} \in \mathbb{R}^{h \times d}$ are the parameters and $x \in \mathbb{R}^d$ is the input. If each element of $\mathbf{W}$ and $\mathbf{W}'$ is sampled from an *continuous probability distribution that is bounded* on $[-1/\sqrt{d}, 1/\sqrt{d}]$, and each element of $\mathbf{v}$ and $\mathbf{v}'$ is sampled from an *continuous probability distribution that is centered and bounded* on $[-1/\sqrt{h}, 1/\sqrt{h}]$, then for any $x \in \mathbb{R}^d$ such that $\|x\|_2 = \sqrt{d}$, with probability $1 - \delta$ over $\mathbf{W}, \mathbf{W}', \mathbf{v}, \mathbf{v}'$, there exist transformations $T, T'$ such that

$$|f_{\{\alpha \tilde{\mathbf{v}} + (1-\alpha)\tilde{\mathbf{v}}', \alpha \bar{\mathbf{W}} + (1-\alpha)\bar{\mathbf{W}}'\}}(x) - \alpha f_{\{\mathbf{v}, \mathbf{W}\}}(x) - (1-\alpha) f_{\{\mathbf{v}', \mathbf{W}'\}}(x)|$$

$$\leq \begin{cases} \tilde{O} \left( \left( \frac{h^2}{(r+r')-h} \right)^{-\frac{1}{2d+4}} \right) & , (r+r') - h > 0 \\ 0 & , \text{otherwise} \end{cases} \tag{28}$$

where $\tilde{\mathbf{v}}, \tilde{\mathbf{W}}$ are transformed versions of $\mathbf{v}, \mathbf{W}$ from $T$ and $\tilde{\mathbf{v}}', \tilde{\mathbf{W}}'$ are transformed versions of $\mathbf{v}', \mathbf{W}'$ from $T'$ respectively. $0 < r, r' \leq h$ are the hidden unit amounts each network can be reduced to via its respective $M, M'$ transformation before being expanded back to width $h$ via $U, U'$.

**Proof.** The proof for Theorem 1.1 is takes a very similar form to Theorem 1, with two differences. For what follows, we assume the same notation and definitions as in Section G.3 up to Eq. (11), with one change: each element of $\mathbb{K}, \mathbb{K}'$ need not be assigned to each $s \in S_\xi$ with equal probability.

Despite this change, for a given $s \in S_\xi$, we can use the Hoeffding's Inequality to bound the size of $C_s(\tilde{\mathbf{W}})$ with high probability:

$$P(|S_n - E[S_n]| \geq t) \leq 2 \exp\left(\frac{-2t^2}{n}\right) \tag{29}$$

$$P(||C_s(\tilde{\mathbf{W}})| - E[|C_s(\tilde{\mathbf{W}})|]| \geq t) \leq 2 \exp\left(\frac{-2t^2}{n}\right) \tag{30}$$

$$\tag{31}$$

Despite $E[|C_s(\tilde{\mathbf{W}})|]$ no longer being equal for each $s$, we can take the union bound (1) over the rows of $\tilde{\mathbf{W}}, \tilde{\mathbf{W}}'$ and (2) for each $s \in S_\xi$ to obtain,

$$||C_s(\tilde{\mathbf{W}})| - |C_s(\tilde{\mathbf{W}}')|| \leq 2\sqrt{\frac{n}{2}\log(12|S_\xi|/\delta)} \tag{32}$$

with probability $1 - \delta/3$. Thus, we achieve the same bound on the size of $I, I'$ as in Section G.3.

The second difference is that $\tilde{\mathbf{v}}$ is no longer sampled uniformly from $[-1/\sqrt{h}, 1/\sqrt{h}]$. However, this does not affect anything because we assume $E[\tilde{\mathbf{v}}] = 0$. Noting these changes, we can follow the derivation in Section G.3 and achieve the same bounds.

### G.5 SIMPLIFYING VARIABLES IN THEOREM 1

Theorem 1 can be simplified to match its introduction in Section 4. First, let $\tau, \tau'$ denote the proportions of features from $h$ that are reduced under the $M, M'$ transformation in each model. By definition, $\tau, \tau' \in [0, 1]$. Then, we can define $r$, and $r'$ in terms of $\tau, \tau'$, and $h$:

$$r = h(1 - \tau), \text{ and } r' = h(1 - \tau')$$

This means $(r + r') - h = h(1 - \tau - \tau') \leq h(1 - 2\min(\tau, \tau'))$. Let $\Gamma = \min(\tau, \tau')$, then we achieve $h(1 - 2\Gamma)$ in the denominator of our bound, which simplifies to what is written in Section 4.

## H EXPERIMENTS IN SETTINGS OF CONCURRENT WORKS

Table 10 shows the results of merging $16\times$ width ResNet20 models trained with cross-entropy loss on the CIAR5+5 task. This setting is equivalent to that of Table 2 from the concurrent work of (Yamada et al., 2023), except for two important differences. First, Yamada et al. (2023) add the REPAIR Jordan et al. (2022) algorithm to each merging method, which significantly improves the performance of each merging algorithm by adding new parameters to the merged model.

Second, Yamada et al. (2023) include merging methods that require training in their Table 2. In contrast, we report the performance of each merging method using its original capabilities (i.e., without REPAIR), and without any training (as it is outside our setting). Thus, all results shown in Table 10 are a lower-bound to what is achievable either with REPAIR, or with training. To make Table 10 as close to Table 2 in (Yamada et al., 2023) as possible, we report "Joint Acc" as the average of each method's logits for the ensemble. To the best of our knowledge, "Joint Acc" is thus the same metric used by (Yamada et al., 2023). Overall, we observe that ZipIt! fully-merged outperforms the nearest baseline by over 20% in "Joint Acc", and zipping up to the classification layers

| Method | FLOPs (G) | Accuracies (%) | | | |
|---|---|---|---|---|---|
| | | Joint | Task A | Task B | Avg |
| Model A | 10.88 | $46.7_{\pm 0.6}$ | $94.4_{\pm 0.8}$ | $18.4_{\pm 3.1}$ | $56.4_{\pm 1.7}$ |
| Model B | 10.88 | $40.2_{\pm 9.6}$ | $22.3_{\pm 6.9}$ | $93.8_{\pm 1.7}$ | $58.0_{\pm 2.9}$ |
| W. Avg | 10.88 | $30.4_{\pm 2.4}$ | $46.2_{\pm 13.4}$ | $47.0_{\pm 9.3}$ | $46.6_{\pm 3.9}$ |
| Git Re-Basin ‡ | 10.88 | $27.3_{\pm 0.5}$ | $46.3_{\pm 1.6}$ | $46.3_{\pm 9.3}$ | $38.9_{\pm 7.7}$ |
| Permute | 10.88 | $45.4_{\pm 5.3}$ | $81.3_{\pm 7.9}$ | $79.5_{\pm 10.0}$ | $80.4_{\pm 3.8}$ |
| **ZipIt!**$_{20/20}$ | 10.88 | $\mathbf{66.0}_{\pm 3.9}$ | $\mathbf{88.0}_{\pm 0.9}$ | $\mathbf{86.7}_{\pm 4.0}$ | $\mathbf{87.4}_{\pm 2.0}$ |
| Ensemble | 21.76 | $80.0_{\pm 2.4}$ | $94.4_{\pm 0.8}$ | $93.8_{\pm 1.7}$ | $94.1_{\pm 0.7}$ |
| **ZipIt!**$_{19/20}$ | 10.89 | $72.0_{\pm 2.5}$ | $90.0_{\pm 1.3}$ | $88.2_{\pm 2.7}$ | $89.1_{\pm 1.7}$ |
| **ZipIt!**$_{13/20}$ | 14.52 | $73.4_{\pm 2.7}$ | $91.5_{\pm 0.9}$ | $89.6_{\pm 3.4}$ | $90.5_{\pm 1.6}$ |
| **ZipIt!**$_{7/20}$ | 18.14 | $75.5_{\pm 2.9}$ | $\mathbf{92.7}_{\pm 1.0}$ | $\mathbf{90.7}_{\pm 3.3}$ | $\mathbf{91.7}_{\pm 1.5}$ |

Table 10: **CIFAR-10 (5+5) Cross Entropy.** ZipIt! vs. baselines using ResNet-20 ($16\times$ width). Merging with ZipIt! up to the last layer (ZipIt!$_{19/20}$) nearly achieves the ensemble "Task Avg." performance with half the FLOPs and vastly outperforms the nearest baseline. Partially merging, brings ZipIt! even closer to the ensemble. ‡ refers to (Ainsworth et al., 2022)

(ZipIt!$_{19/20}$) *nearly matches* the ensemble "Task Avg." accuracy without requiring *any training*. Interestingly, Git Re-Basin performs especially poorly in this setting, likely requiring REPAIR to achieve the performance reported in Table 2 by (Yamada et al., 2023).

