# OpenReview forum: "ZipIt! Merging Models from Different Tasks without Training"
_ICLR.cc/2024/Conference — ICLR 2024 poster_

### Official Review · Reviewer_SEiC · 2023-10-12

**Soundness:** 3 good
**Presentation:** 4 excellent
**Contribution:** 2 fair
**Rating:** 6
**Confidence:** 4

**Summary:**

This paper tackles the problem of merging two models trained on different "tasks". Their best performing approach is two-fold: 1) identify redundant features within/across models and average those, and 2) only merge some layers. Combined they achieve performance which approaches the ensemble, but their improvements are most notable when the two "tasks" come from the same task, such as partitioning CIFAR in two parts.

**Strengths:**

- The presentation, including all Figures, is very clear!
- The idea of merging redundant features is interesting and effective. It seems to me that the authors should consider exploring this as a general way of speeding up inference and not just as a technique for merging models.

**Weaknesses:**

- I would appreciate more motivation for this problem. When would I find myself training from scratch on two different tasks?
- [Subjective] I suggest the authors are much more clear/transparent about their contributions early, because currently the readers get their hopes up a lot when in reality the best performance occurs either when the "tasks" really come from the same task like splitting CIFAR in half. I think there's still a long way to go for training individually on two distinct datasets that are ImageNet level difficulty.
- Figure 1a/b is not exactly clear with respect to related work, for instance [1] (Sec 3.3), and [2] (Appendix E) do consider merging models trained on different tasks, albeit from the same initialization.

[1] Merging Models with Fisher-Weighted Averaging (https://arxiv.org/abs/2111.09832)
[2] Model soups: averaging weights of multiple fine-tuned models improves accuracy without increasing inference time (https://arxiv.org/abs/2203.05482)

**Questions:**

- I would be curious when the authors believe this approach is best applied, e.g., when would I find myself training from scratch on two different tasks?
- Does this approach improve accuracy in more traditional model merging settings? It's pitched for multi-task but is it more general?
- The "scaling" trend in Figure 6b is very interesting! I would be curious to hear if the authors would expect the same trend with a different method to increase FLOPs on the x-axis, such as depth scaling, or joint scaling, or training iterations.

---

> ### Author Response · Authors · 2023-11-17
> **Response to Reviewer SEiC**
>
> We thank you for your thoughtful review and suggestions. We have collected our response below.
>
> **W1 / Q1. What is the motivation for this? When would I find myself training from scratch on two different tasks?**
> This is a good question. The assumption that we and prior work make in this setting isn’t that you would train these models yourself, but that they would be given to you (e.g., downloaded from the internet), or trained in a federated learning setting where due to, e.g., privacy reasons, not all the data is available at once. In those cases, we would like to be able to merge the models without any additional training, in the former because it may be too expensive, and in the latter because we are not allowed to access the original data.
>
> We only train these models ourselves to explicitly control the evaluation setting. Note that some of the models in the multi-dataset experiment were not trained by us.
>
> **W2. More transparent claims.**
> Thank you for this comment. We’ve updated the draft to tone down the language in our PDF abstract and Fig 1 caption.
>
> **W3. Figure 1a/b is not exactly clear with respect to related work.**
> We appreciate the reviewer for pointing this out and we completely agree. We have updated our draft to specify methods with _different initializations_ in the caption of that figure.
>
> **Q2. Does this improve on same data merging settings?**
> We match the performance of baselines on the same data settings, but we focus on the multi-dataset case.
>
> **Q3. I would be curious to hear if the authors would expect the same trend with a different method to increase FLOPs on the x-axis, such as depth scaling, or joint scaling, or training iterations.**
> We did not conduct this search, but it is unlikely that we would see the same magnitude of gains as from scaling width. Mode connectivity heavily relies on the width of models, not necessarily the depth or other possible scaling components. Though, ZipIt is less constrained because of Theorem 1, so it is likely that it would still outperform the other methods in those cases.

---

> > ### Comment · Reviewer_SEiC · 2023-11-21
> >
> > Thank you for your response.

---

### Official Review · Reviewer_GcCE · 2023-10-28

**Soundness:** 2 fair
**Presentation:** 3 good
**Contribution:** 3 good
**Rating:** 6
**Confidence:** 3

**Summary:**

This paper proposes a method for merging trained models trained on different datasets without additional training based on feature correlations. The proposed method can further adjust the inference cost and accuracy by adjusting the number of layers to be merged.

**Strengths:**

1. This paper addresses the interesting research topic of merging models on different datasets without additional training.
2. Large-scale experiments such as imagenet1k confirm the effectiveness of the proposed method. This paper conducts experiments in multi-dataset merging, classification and segmentation merging. In particular, this is the first effort, to my knowledge, to merge the different modalities of classification and segmentation.
3. For reproducibility, the authors mentioned that they plan to publish the source code; the cost of replication experiments would go down significantly if the checkpoints of the trained models were made public since they involve large scale experiments such as cifar100, imagenet1k, and multidataset.

**Weaknesses:**

1. The paper extends feature matching between models to within the model in order to merging models across different datasets. Its contribution should be examined in more detail. In particular, there is no theoretical support as to why matching the feature would lower the loss after merging. I did not understand the connection between Theorem 1 and the fact that joint Acc can be sufficiently large for merging models on different datasets. Is the extension to "within merge" not only more flexible in alignment, but also suitable for merging models on different datasets?
2. The proposed method is inaccurate for meging of imagenet1k and multi dataset unless partial zip is used, while the inference cost increases as the number of models increases with partial zip.

**Questions:**

1. Is git-rebasin activation matching considered equivalent to zipit without wighin merge? I would like to know the difference between git rebasin, permute and zipit.
 2. Does merge by zipit keep the output strictly invariant like permutation, even when using pseudoinverse matrix?
 3. What is the same model budget and explain how beta is introduced as an equation?
 4. Three matching methods were proposed in git-rebasin: activation matching, weight matching, and STE. Which method is the baseline in Table 1 and 2? I doubt the claim that permutation alone makes it difficult to merge models on different datasets. According to [1], using STE, which is a permutation base, and Merge models with cifar10(5+5), it is reported that 90% accuracy is achieved, which is 10% more accurate than zipit. I think the conditions are the same as zipit!20/20 in terms of combining all layers.
 5. Sec 5.1 says " `If allow the last stage of the network to remain unzipped (i.e., zip up to 13 layers), our method obtains 83.8%, which is only 3.6% behind an ensemble of model A and model B (which is practically the upper bound for this setting)`", but how can ensemble be guaranteed to be uppper bound?  In [1], a merging model is reported that can achieve higher accuracy than an ensemble. The upper limit of joint accuracy is the accuracy of the model trained on all data, rather than the accuracy of the ensemble.
 6. Why is partially zipping more accurate than fully zipping?
 7. How much gpu is needed for Imagenet-1k (200+200)? I am concerned about the memory and computational cost required to compute the correlation matrix.
[1]: https://arxiv.org/abs/2306.05641




===========================================

Comments after reading the rebuttal.

===========================================

Since I was unable to set the readers to everyone in the reply, I am writing my comments here.

Thank you for your kind reply.
I did not understand that merging models using loss like STE is outside the scope of this study.
Since this has been resolved in regards to many of my questions, I would like to raise my score.

As other reviwers have pointed out, it would be helpful to specify as a limitation in the camera-ready version of the paper that the model must be wide enough to perform.

---

> ### Author Response · Authors · 2023-11-17
> **Response to Reviewer GcCE (1/2)**
>
> We thank you for your review and feedback. We would like to clarify a couple points about our setting and the comparisons to Git-Rebasin:
>
> **Note 1. Our Setting.**
> To clarify, we work in the same setting of [2,4,5], where the user would like to merge two models that were trained independently (i.e., different initialization) without full access to the original data and without having to train—with the added stipulation of the models being trained on different tasks. To the best of our knowledge, we are the first to tackle the setting from this angle. Solving this task would allow us to, for instance, merge foundation models for which we do not have the compute to train, or improve federated learning techniques.
>
> In this setting, the ensemble is the true upper bound, as training is not allowed so the perfect merging would exactly match the output of the original models. Additionally, the extension of working on different tasks / datasets _broadens_ the possible uses of ZipIt! beyond these prior work. Allowing training (like for instance, STE in Git-Rebasin) would increase our performance further, but is out of scope for this work.
>
> **Note 2. Which version of Git-Rebasin?**
> We conduct all our Git-Rebasin [2] experiments in their official public code release [6], which for all ResNet models employs _weight-matching (WM)_. Note that [2] find weight-matching actually performs competitively (and better) to their activation-matching version, though our permute baseline performs better than both. We’ve updated our draft to mention that we use the weight-matching version.
>
> **Note 3. The Permute baseline.**
> We noticed there may be a misunderstanding about this baseline. The permute method is an ablation of ZipIt!. We use our algorithm, but without merging-within, and utilizes the optimal Hungarian algorithm to find permutations. Thus, it is a strong baseline that already outperforms prior work.
>
> We have also included our response to each of the reviewer’s concerns below.
>
> **W1. My understanding is that Theorem 1 is positioned to support merging on different datasets + within merging. But this doesn’t make sense to me.**
> We clarify that we only position Theorem 1 to motivate why merging **within** models _is better_ than permutation-based methods when redundancy exists within models. We agree that Theorem 1 does not explicitly provide justification for why we would expect strong performance when using within-merging to merge models on different tasks. Instead, we demonstrate this empirically through our extensive experiments (Tables 1/2/3, Figure 6b in the main paper, and more in the Appendix).
>
> **W2. Merged model is inaccurate on imagenet1k and multi-dataset unless partial zip is used.**
> It is correct that ZipIt! does not achieve the same performance on ImageNet1k as it does on CIFAR. Yet, we note that _all our baselines including git-rebasin (WM) perform significantly worse_ than ZipIt!, illustrating the task difficulty when **retraining is not allowed (note 1)**. However, we attribute this result to model capacity. It is likely that our ResNet50 is not large enough to contain sufficient redundancy for which ZipIt! (or the baselines / prior work) can make use of. Based on Figure 6b, we expect performance to dramatically improve with scale.
>
> For low-capacity models, partial zipping is necessary to avoid merging features in later layers that have very low correlation with one another, thus avoiding catastrophic performance when not allowed to train. See our response to Q6 for more details.
>
> **Q1. Is git-rebasin activation matching considered equivalent to zipit without within merge? I would like to know the difference between git rebasin, permute and zipit.**
> ZipIt! within-merging utilizes our _own algorithm_ which achieves near optimal performance (Table 4) whilst being significantly faster and more scalable. The permute method is an ablation of ZipIt!. We use our algorithm, but without merging-within, and utilizes the Hungarian algorithm to find permutations. Git-rebasin with activation matching (AM) is _different_ from both these methods. AM finds permutations based on the pairwise dot-products of neuron activations between models and it appears worse than weight-matching (WM) (Figure 2 of [2]). However, both our permute and ZipIt! significantly outperform WM in every experiment.
>
> Notably, ZipIt! is the only method capable of jointly aligning >2 models into a shared space, as opposed to aligning all models to the space of a chosen anchor model. All prior work (including Git-Rebasin) does the latter, and it is very sensitive to the choice of anchor model.

---

> > ### Author Response · Authors · 2023-11-17
> > **Response to Reviewer GcCE (2/2)**
> >
> > **Q2. Does merge by zipit keep the output strictly invariant like permutation, even when using pseudoinverse matrix?**
> > Permuting doesn’t change the output of the model, but adding two permuted models together _does_ change the output. In ZipIt!, we do both the transforming and adding in one step, so the output isn’t invariant at any point unless we set the within merge budget to 0 (See Q3). However, we consistently demonstrate that ZipIt! outperforms function preserving in-alignment baselines like permute.
> >
> > **Q3. What is the same model budget and explain how beta is introduced as an equation?**
> > We apologize for our lack of clarity here. “Same model budget” refers to the proportion of merges that we allow to be _within the model_, as opposed to across as in prior work. A budget of 0 indicates no merging-within is allowed, while a budget of 1 allows ZipIt! to merge all the features _within_.
> >
> > For implementation, we define the within model merging budget for $m$ models as $\text{budget}_i = n\beta / m$ where $n$ is the number of features for a single model and each model shares the same budget (i.e., $1/m$ of the overall). Then, whenever a within merge is selected, we decrement that count for that model. Once the counts reach zero, we do not allow any within merges for that model.
> >
> > **Q4. Three matching methods were proposed in git-rebasin: activation matching, weight matching, and STE. Which method is the baseline in Table 1 and 2? I doubt the claim that permutation alone makes it difficult to merge models on different datasets. According to [1], using STE, which is a permutation base, and Merge models with cifar10(5+5), it is reported that 90% accuracy is achieved, which is 10% more accurate than zipit. I think the conditions are the same as zipit!20/20 in terms of combining all layers.**
> > Thank you for bringing this to our attention. We compare against the weight matching version, as that is the default mode that Git Rebasin uses in all of its experiments. Note that the STE version of Git Rebasin is not a valid comparison to our method, as STE requires training on the joint dataset. As mentioned before, we specifically do not allow training after or during merging in our setting. Moreover, the numbers themselves are not comparable because [1] uses much wider models (x16 vs our x4) as well as REPAIR [4] after the fact to increase accuracy, (which we could use, but omit to avoid adding extra parameters to the model). We’ve added Appendix H in the updated draft to explain these differences.
> >
> >
> > **Q5. Why is the ensemble the upper bound? Shouldn’t it be joint accuracy?**
> > This is because without being able to retrain, a perfect merge would be when the merged model exactly outputs what the original two models output—which is the ensemble performance. If training is allowed (like the STE method you mention in [2]), then the upper bound becomes jointly trained accuracy as you state. However, our setting doesn’t allow training. We’ve clarified this in the baselines section of the updated draft.
> >
> > **Q6. Why is partially zipping more accurate than fully zipping?**
> > The average correlation between features are much higher in the early layers (corroborated by [3]). Here are the average correlations for each stage in a ResNet-20x8 on CIFAR 50+50:
> >
> > | Layer 7/20 | Layer 13/20 | Layer 19/20 |
> > |:----------:|:-----------:|:-----------:|
> > |    0.50±0.01    |     0.37±0.00    |     0.27±0.00    |
> >
> > In these cases, forcibly merging unrelated (i.e., low correlations) features together destroys their information and results in very poor merges, which is why partial zipping leads to higher accuracy. We’ve included this table and explanation in Appendix A of the updated draft.
> >
> > **Q7. How much gpu is needed for Imagenet-1k (200+200)? I am concerned about the memory and computational cost required to compute the correlation matrix.**
> > The total GPU memory overhead is very low. Specifically, we compute the correlation matrix on the CPU with an efficient iterative algorithm that takes just a few seconds.
> >
> >
> >
> >
> >
> > [1] https://arxiv.org/abs/2306.05641
> > [2] GitRebasin: https://arxiv.org/pdf/2209.04836.pdf
> > [3] Simon Kornblith, Mohammad Norouzi, Honglak Lee, and Geoffrey Hinton. Similarity of neural network representations revisited. In ICML, 2019.
> > [4] REPAIR: https://arxiv.org/pdf/2211.08403.pdf
> > [5] OT-Fusion: https://arxiv.org/pdf/1910.05653.pdf
> > [6] Git-Rebasin Code: https://github.com/samuela/git-re-basin

---

### Official Review · Reviewer_pbds · 2023-10-30

**Soundness:** 3 good
**Presentation:** 2 fair
**Contribution:** 2 fair
**Rating:** 5
**Confidence:** 4

**Summary:**

This paper proposes a novel method for merging two different models trained from different tasks, maintaining the performance on both tasks. The method resolves the problem of decreasing performance on two disjoint tasks, where prior works fail, ﻿which ﻿permutes the model to another one and the average of them. Moreover, this paper proposed partially zipping to get the trade-off between the performance and the computational complexity.

**Strengths:**

This paper proposes a novel method named “ZipIt!” which can maintain the performance after merging two models trained from two disjoint tasks, which prior works fail. The partially zipping method can further give the option to do a trade-off between the performance and the FLOPs. Finally, a theoretical proof is given to ensure the existence of a transformation matrix.

**Weaknesses:**

The method seems not to be practical for real-world usage, while lacks insights and theoretical analysis of the model properties. The method works fine in the CIFAR dataset, but doesn’t have remarkable advantages in large datasets such as ImageNet, and disjoint tasks. Moreover, the paper lacks comparisons between its method and the model trained on both datasets directly, which limits its usage scenario.

**Questions:**

- I’ve noticed that the “permute” baseline is stronger than “Git Re-Basin”, which is a little strange. I’m wondering how it happens and what’s the experimental setting?
- The improvement on CIFAR is remarkable, while in the larger dataset ImageNet, the trade-off between the performance and FLOPs is debatable. Compared to the ensemble, the FLOPs don’t decrease a lot while getting lower accuracy.
- I’m curious about the comparison between “ZipIt!” method and a single model trained on both datasets. Did you think about this?
- The presentation for “Ensemble” results which uses light grey is a little misleading.
- Are there the results of Git Re-Basin method for ﻿Multi-Dataset in Table 3?
- Did you consider the time cost of computing transformation matrices in your method when comparing the FLOPs?

---

> ### Author Response · Authors · 2023-11-17
> **Response to Reviewer pbds**
>
> We thank you for your review and feedback. We have responded to your concerns below.
>
> **Clarification: The Permute baseline.**
> We noticed there may be a misunderstanding about this baseline. The permute method is an ablation of ZipIt!. We use our algorithm, but without merging-within, and utilize the optimal Hungarian algorithm to find permutations. Thus, it is a strong baseline that already outperforms prior work.
>
> **W1. “The method seems not to be practical for real-world usage, while lacks insights and theoretical analysis of the model properties”.**
> Our main goal is to show that it is possible to obtain good performance by merging models from different tasks **without retraining**, which is something no prior work has been able to achieve. Thus, to show this we present merged accuracy in many different settings with many more evaluations in the appendix. While a theoretical analysis of the resulting models might be interesting, it is not the core focus of our work.
>
> However, we do include extensive analysis of different aspects of the resulting models in Fig. 6/7/8 and Sec. 6, as well as provide rigorous theoretical motivation for our method in Eq. 4 and Appendix G.
>
>
> **W2. “The method works fine in the CIFAR dataset, but doesn’t have remarkable advantages in large datasets such as ImageNet, and disjoint tasks.”**
> We would like to emphasize that ZipIt! _significantly_ outperforms prior work on both CIFAR-10 and CIFAR-100 and gets close to the _maximum possible_ performance in these settings (i.e., ensemble accuracy). For much harder settings like ImageNet, ZipIt! still outperforms prior work and is able to perform significantly better when using partial unzipping. Note that the “permute baseline” is a **strong baseline** we introduce that _already_ outperforms prior work.
>
>
> **W3. Moreover, the paper lacks comparisons between its method and the model trained on both datasets directly, which limits its usage scenario.**
> To clarify, we work in the same setting of [1,2,3], where the user would like to merge two models that were trained independently (i.e., different initialization) without full access to the original data and without having to train—with the added stipulation of the models being trained on different tasks. To the best of our knowledge, we are the first to tackle the setting from this angle. Solving this task would allow us to, for instance, merge foundation models for which we do not have the compute to train, or improve federated learning techniques.
>
> In this setting, the ensemble is the true upper bound, as training is not allowed, so the perfect merging would exactly match the output of the original models. Additionally, the extension of working on different tasks / datasets _broadens_ the possible uses of ZipIt! beyond these prior work. Allowing training would increase our performance further, but is out of scope for this work.
>
>
> **Q1. Permute stronger than Git Rebasin?**
> We conducted all our Git-Rebasin experiments in their official public code release [4], which for all ResNet models employs _weight-matching_. Note that [1] find weight-matching actually performs competitively (and better) to the activation-matching version. Our permute baseline is simply our ZipIt! algorithm with the matching algorithm replaced by an optimal permutation finding algorithm. Thus, it is a _strong_ baseline and already outperforms prior work.
>
> **Q2. I’m curious about the comparison between “ZipIt!” method and a single model trained on both datasets.**
> See our response for W3 above. While it might be an interesting comparison, this is out of scope for our setting as the upper bound is ensemble performance.
>
> **Q3. The presentation for “Ensemble” results which uses light grey is a little misleading.**
> Thank you for your comment. We gray out the ensemble because it requires evaluating both models, which uses significantly more (up to double) the parameters / compute compared to the rest of the table entries. Do you have any suggestions on how we can make this more clear?
>
> **Q4. Are there the results of Git Re-Basin method for ﻿Multi-Dataset in Table 3?**
> Git Rebasin in its original code-base (all Git-Rebasin experiments are conducted in the official repository [4]) does not support this type of experiment. However, we expect its performance to be much worse than our permute method just like in Tab. 1 and Tab. 2.
>
> **Q5. Did you consider the time cost of computing transformation matrices in your method when comparing the FLOPs?**
> The transformation matrices only need to be computed once and are very quick to compute (~1s total on a CPU), so it is not included in the model FLOPs. Benchmark results for the actual transformation calculation can be found in Tab. 4.
>
>
> [1] Git-Rebasin: https://arxiv.org/pdf/2209.04836.pdf
> [2] OT-Fusion: https://arxiv.org/pdf/1910.05653.pdf
> [3] REPAIR: https://arxiv.org/pdf/2211.08403.pdf
> [4] Git-Rebasin Code: https://github.com/samuela/git-re-basin

---

> ### Comment · Reviewer_pbds · 2023-12-04
>
> Thanks for the response. However, the authors have not addressed this problem that "the method seems not to be practical" as "compared to the ensemble, the FLOPs don’t decrease a lot while getting lower accuracy". This indicates that “ZipIt!” may not achieve a good trade-off between performance and training time. Therefore, I would like to maintain this 5 score.

---

### Official Review · Reviewer_3tA9 · 2023-11-01

**Soundness:** 3 good
**Presentation:** 4 excellent
**Contribution:** 3 good
**Rating:** 6
**Confidence:** 3

**Summary:**

The paper proposes a method to merge two different models sharing the same architecture, possibly with different initialization and for different tasks, without re-training. The key idea is to merge the weights whose outputs are highly correlated, both within and across models. The method also allows weights to be partially merged, adding flexibility to adjust the amount of merging depending on resource budget and accuracy requirements. Experiments show that ZipIt! outperforms the existing baselines in several settings, including merging classification models and merging across tasks.

**Strengths:**

1. This paper studies a new and interesting problem: merging differently initialized models trained on different tasks into a single model without retraining. This relaxes the condition of the model merging problem, recently studied in works such as Model Soups and Git Re-Basin, which assumes the same task over models. This makes the problem more challenging while allowing for broader use cases.

2. The idea of merging weights both within and across models is simple, but achieves reasonably good performance under different conditions in the experiments. It's great that experiments with merging tasks of different modalities (classification, segmentation) are also done and the method still achieves reasonable performances, showing that the method is generalizable to non-classification tasks. In addition, the ability to flexibly adjust the amount of merging is also a strong point.

3. The paper is well written. It was easy to follow. It is also great that the code is already publicly available.

**Weaknesses:**

1. My main concern is the stability of the method. I am not sure if the method works robustly across different settings, because empirical observations are reported in the github issues that a small change (e.g. changing the learning rate and epoch numbers when training the models to merge) can cause the merged model to crash to low accuracy.

2. These are not weaknesses but there are some limitations. The performance drop increase when merging multiple models. Accuracy drop looks large when merging models of different tasks (Table 8). All experiments used CNNs and transformers are not tested.

**Questions:**

1. The meaning of "ensemble" in the experiments is unclear to me. I don't think I understood how it works.

2. I would appreciate more experiments to test the stability of the method. It would also be nice to provide an analysis of when the method works and when it does not. Please see the weakness section for more details.

3. It would also be interesting to include additional experiments on more compact architectures such as Mobilenet for a more comprehensive analysis of the behavior of the method. It is great that ZipIt! is already evaluated on models with larger widths (x1.5 in Table 7). I think it might be harder to merge models where the architectures are already compact because there might be less overlap and less redundancy of features.

4. Can BN be removed before applying ZipIt by fusing it into Conv (using that BN is an affine transform when mean and variance are fixed)? I think then the process to reset BN after merge can be avoided.

---

> ### Author Response · Authors · 2023-11-17
> **Response to Reviewer 3tA9**
>
> We thank you for your comprehensive review and feedback. We have included responses to your questions and concerns below.
>
> **W1 / Q2. Concern about Stability.**
> Most mode connectivity and model merging work assumes models are well-trained. In the issue referenced, the original models obtained 15% lower accuracy than well-trained models, as they were underfit with too low of a learning rate, and thus themselves underperformed before the merge. This caused all merging algorithms to drop to under 20% in accuracy, so this is not specifically a ZipIt! issue. The merged model’s performance is naturally going to depend on the original models’ performances.
>
> **W2. Performance degradation when merging multiple models. What about transformers?**
> Unfortunately, that is a limitation of merging multiple models trained on different tasks into the size of one without being able to train–-since each model has some task-specific neurons [1]. However, as shown in Fig. 6b, this could likely be alleviated in practice by increasing the network width. As for transformers, at the time of submission no other model merging work has shown results on transformers with different initializations, but this is a great direction for future work.
>
> **Q1. What does “ensemble” mean here?**
> Thanks for pointing this out, we’ve clarified this in the baseline section of the updated draft. For joint accuracy, we softmax the logits of each model separately to place their predictions on the same scale, and then concatenate resulting probabilities from each model and take an argmax. For avg task accuracy, we use the correct model for each task (i.e., just average the per-task accuracies). Note that this is upper bound accuracy because with perfect merging (not allowing training), the merged model's outputs would be identical to the originals.
>
> **Q3. Results for smaller models, e.g., MobileNet?**
> For smaller models, note that ResNet-20x4 (which we use for our CIFAR experiments) has around the same number of parameters as MobileNetV2, and we show scaling performance (both downward and upward) in Fig. 6b. While it’s true that constraining the model size hurts performance, this depends on task difficulty and is true for all other methods as well.
>
> **Q4. Can BN be fused beforehand, so it doesn’t need to be recomputed?**
> Sadly, while it is possible to fuse the BNs into the models beforehand, [2] shows that simply merging the model’s BNs together would result in “variance decay”. So some mitigation strategy would be necessary regardless, and we found resetting the batch norm to be the simplest option.
>
> [1] Simon Kornblith, Mohammad Norouzi, Honglak Lee, and Geoffrey Hinton. Similarity of neural network representations revisited. In ICML, 2019.
> [2] REPAIR: https://arxiv.org/pdf/2211.08403.pdf

---

### Meta-Review · Area_Chair_rYaV · 2023-12-08

**Metareview:**

This paper tackles model merging in the challenging setting where the models-to-be-merged do not necessarily share an initialization but do share an architecture. There are two primary innovations: The first is to find the superset of features across the two models and ensure all of them remain in the merged model, and the second is to only merge the model up to a certain layer and keep the rest of the layers fixed. Both of these operations are "relaxations" of the classical merging setup in the sense that merging typically involves creating a single model that is no larger than the constituent models. This paper, however, provides compelling arguments that this is beneficial. It also builds naturally on merging methods that aim to account for permutation symmetries. Ultimately, on the task of merging vision models, the proposed framework improves performance drastically.

**Justification For Why Not Higher Score:**

While the gains in this paper are impressive, a major source of these gains is a relaxation of the merging problem where the merged model can have more parameters/be more computationally expensive than the constituent models. This paper proposes a good solution to this new problem setting, and I think it will motivate future work in the area.

**Justification For Why Not Lower Score:**

Consensus was generally to accept.

---

### Decision · Program_Chairs · 2024-01-16

Accept (poster)